# FGF21 protects against hepatic lipotoxicity and macrophage activation to attenuate fibrogenesis in nonalcoholic steatohepatitis

Cong Liu[1,2], Milena Schönke[1,2], Borah Spoorenberg[1,2], Joost M Lambooij[3,4], Hendrik JP van der Zande[3], Enchen Zhou[1,2], Maarten E Tushuizen[5], Anne-Christine Andreasson[6], Andrew Park[7], Stephanie Oldham[8], Martin Uhrbom[6], Ingela Ahlstedt[6], Yasuhiro Ikeda[7], Kristina Wallenius[6], Xiao-Rong Peng[6], Bruno Guigas[3], Mariëtte R Boon[1,2]*, Yanan Wang[9]*, Patrick CN Rensen[1,2,9]*

[1]Department of Medicine, Division of Endocrinology, Leiden University Medical Center, Leiden, Netherlands; [2]Einthoven Laboratory for Experimental Vascular Medicine, Leiden University Medical Center, Leiden, Netherlands; [3]Department of Parasitology, Leiden University Medical Center, Leiden, Netherlands; [4]Department of Cell and Chemical Biology, Leiden University Medical Center, Leiden, Netherlands; [5]Department of Gastroenterology and Hepatology, Leiden University Medical Center, Leiden, Netherlands; [6]Bioscience Metabolism, Research and Early Development, Cardiovascular, Renal and Metabolism (CVRM), BioPharmaceuticals R&D, AstraZeneca, Gothenburg, Sweden; [7]Biologics Engineering and Targeted Delivery, Oncology R&D, AstraZeneca, Gaithersburg, United States; [8]Bioscience Metabolism, Research and Early Development, Cardiovascular, Renal and Metabolism (CVRM), BioPharmaceuticals R&D, AstraZeneca, Gaithersburg, United States; [9]Med-X institute, Center for Immunological and Metabolic Diseases, and Department of Endocrinology, First Affiliated Hospital of Xi'an Jiaotong University, Xi'an Jiaotong University, Xi'an, China

*For correspondence:
m.r.boon@lumc.nl (MRB);
y_wang@xjtufh.edu.cn (YW);
p.c.n.rensen@lumc.nl (PCNR)

**Abstract** Analogues of the hepatokine fibroblast growth factor 21 (FGF21) are in clinical development for type 2 diabetes and nonalcoholic steatohepatitis (NASH) treatment. Although their glucose-lowering and insulin-sensitizing effects have been largely unraveled, the mechanisms by which they alleviate liver injury have only been scarcely addressed. Here, we aimed to unveil the mechanisms underlying the protective effects of FGF21 on NASH using APOE*3-Leiden.CETP mice, a well-established model for human-like metabolic diseases. Liver-specific FGF21 overexpression was achieved in mice, followed by administration of a high-fat high-cholesterol diet for 23 weeks. FGF21 prevented hepatic lipotoxicity, accompanied by activation of thermogenic tissues and attenuation of adipose tissue inflammation, improvement of hyperglycemia and hypertriglyceridemia, and upregulation of hepatic programs involved in fatty acid oxidation and cholesterol removal. Furthermore, FGF21 inhibited hepatic inflammation, as evidenced by reduced Kupffer cell (KC) activation, diminished monocyte infiltration, and lowered accumulation of monocyte-derived macrophages. Moreover, FGF21 decreased lipid- and scar-associated macrophages, which correlated with less hepatic fibrosis as demonstrated by reduced collagen accumulation. Collectively, hepatic FGF21 overexpression limits hepatic lipotoxicity, inflammation, and fibrogenesis. Mechanistically, FGF21 blocks hepatic lipid influx and accumulation through combined endocrine and autocrine signaling, respectively, which prevents

KC activation and lowers the presence of lipid- and scar-associated macrophages to inhibit fibrogenesis.

## Editor's evaluation

The study examines the mechanism of hepatic FGF21 using trangenic and over-expression models to show that it limits hepatic lipotoxicity, inflammation and fibrogenesis. They provide convincing data to support the notion that FGF21 blocks hepatic lipid influx and accumulation through combined endocrine and autocrine signaling, respectively, which prevent Kupffer cell activation, and scar-associated macrophages to inhibit fibrogenesis.

## Introduction

The liver is the nexus of many metabolic pathways, including those of glucose, fatty acids (FAs), and cholesterol. In health, these metabolites are distributed to peripheral tissues while preventing long-lasting accumulation in the liver. In a pathological state, however, lipids may accrue in the liver, thereby impairing liver function and carving the path toward the development of nonalcoholic fatty liver disease (NAFLD) (*Cusi, 2012*). NAFLD is considered a spectrum of liver diseases ranging from liver steatosis, characterized by lipid accumulation in hepatocytes, to nonalcoholic steatohepatitis (NASH) with hepatic steatosis, lobular inflammation, hepatocyte ballooning, and varying degrees of fibrosis (*Friedman et al., 2018*; *Arab et al., 2018*). Patients diagnosed with NASH are predisposed to developing cirrhosis and hepatocellular carcinoma, among whom patients with severe liver fibrosis are at greatest risk of overall- and liver-related mortality (*Taylor et al., 2020*). Despite this, there are currently no approved pharmaceutical therapeutics for NASH. Instead, lifestyle modifications remain the first-line treatment for NASH, although this is rarely attainable in the long term, and liver transplantation is still the sole intervention to treat the end-stage of NASH (*Friedman et al., 2018*; *Stefan et al., 2019*). Thus, there is an unmet need for therapeutic strategies that control the progression of NASH, in particular of liver fibrosis, and reverse the underlying pathophysiology.

Current hypotheses suggest that adipose tissue dysfunction and lipid spillover leads to hepatic lipotoxicity, and thereby the initiation of NASH (*Musso et al., 2009*; *Neuschwander-Tetri, 2010*), which further progresses through the inflammatory response triggered by hepatic lipotoxicity (*Neuschwander-Tetri, 2010*). This inflammatory response and subsequent fibrogenesis are primarily initiated by liver macrophages (*Tacke, 2017*). Hepatic macrophages mainly consist of embryonically derived macrophages, termed resident Kupffer cells (ResKCs), and monocyte-derived macrophages (MoDMacs) that are recruited from the circulation (*Krenkel and Tacke, 2017*). In the steady state, ResKCs serve as sentinels for liver homeostasis. In NASH, liver injury caused by excess lipids and hepatocyte damage/death triggers ResKC activation, leading to pro-inflammatory cytokine and chemokine release (*Tran et al., 2020*). This fosters the infiltration of newly recruited monocytes into the liver, which gives rise to various pro-inflammatory and pro-fibrotic macrophage subsets (*Tacke, 2017*; *Tran et al., 2020*). Interestingly, recent preclinical and clinical studies have reported that modulation of ResKC activation, monocyte recruitment, or macrophage differentiation, to some extent, can attenuate NASH (*Tacke, 2017*; *Krenkel et al., 2018*). In light of these findings, FGF21, a hepatokine with both lipid-lowering and anti-inflammatory properties (*Meng et al., 2021*; *Guo et al., 2016*), has been brought to the foreground as a promising potential therapeutic to treat NASH.

The specificity of FGF21 action for various metabolic tissues is determined by the FGF receptor (FGFR) which forms a heterodimer with the transmembrane co-receptor β-Klotho (KLB) (*Fisher and Maratos-Flier, 2016*; *Geng et al., 2020*). While the FGFR is ubiquitously expressed, KLB is primarily expressed in the liver and adipose tissue (*Fisher and Maratos-Flier, 2016*; *Geng et al., 2020*), therefore possibly limiting FGF21 action to these tissues. Physiologically, FGF21 is considered a stress-induced hormone whose levels rise in metabolically compromised states, such as obesity (*Zhang et al., 2008*) and NASH (*Barb et al., 2019*). The increased FGF21 in these pathologies is likely induced by an accumulation of lipids in the liver (*Li et al., 2010*). As such, plasma FGF21 also positively correlates with the severity of steatohepatitis and fibrosis in patients with NASH (*Barb et al., 2019*). Induction of FGF21 is thought to mediate a compensatory response to limit metabolic dysregulation (*Flippo and Potthoff, 2021*), although this level is not sufficient. Interestingly, two-phase 2a clinical trials reported

**eLife digest** High-calorie modern diets have contributed to growing rates of obesity-linked diseases. One such disease is non-alcoholic steatohepatitis or NASH for short, which affects about 5% of adults in the United States. The livers of people with this condition accumulate fat, become inflamed, and develop scar tissue. People with NASH are also at increased risk of developing liver cancer, type 2 diabetes, and heart disease. Currently, no drugs are available to treat the condition and prevent such severe complications.

Previous research has shown the liver produces a stress hormone, called FGF21, in response to fat accumulation. This hormone boosts fat burning and so helps to reduce excess fat in the liver. Drugs that mimic FGF21 have already been developed for type 2 diabetes. But so far, it was unclear if such drugs could also help reduce liver inflammation and scarring in patients with NASH.

Liu et al. show that increasing the production of FGF21 in mice with a NASH-like condition reduces fat accumulation, liver inflammation, and scarring. In the experiments, the researchers used gene therapy to ramp up FGF21 production in the livers of mice that develop obesity and a NASH-like condition when fed a high-fat diet for 23 weeks. Increasing FGF21 production prevented the mice from developing obesity while on the high fat diet by making the body burn more fat in the liver and brown fat tissue. The treatment also reduced inflammation and prevented scarring by reducing the number and activity of immune cells in the liver.

Increasing the production of the stress hormone FGF21 prevents diet-induced obesity and NASH in mice fed a high-fat diet. More studies are necessary to determine if using gene therapy to increase FGF21 may also cause weight loss and could reverse liver damage in mice that already have NASH. If this approach is effective in mice, it may be tested in humans, a process that may take several years. If human studies are successful, FGF21-boosting therapy might provide a new treatment approach for obesity or NASH.

that pharmacological FGF21 treatment improves liver steatosis in NASH patients (*Sanyal et al., 2019*; *Harrison et al., 2021*). While an in vivo study testing the therapeutic potency of FGF21 in choline-deficient and high-fat diet-induced NASH has previously reported both anti-inflammatory and anti-fibrotic effects (*Bao et al., 2018*), detailed mechanistic understanding is still lacking.

In the present study, we aimed to elucidate the mechanisms underlying FGF21-mediated improvement of NASH, in particular of steatohepatitis and fibrogenesis. To this end, we used APOE*3-Leiden. CETP mice, a well-established model for human cardiometabolic diseases. These mice exhibit human-like lipoprotein metabolism, develop hyperlipidemia, obesity, and inflammation when fed a high-fat high-cholesterol diet (HFCD), and develop fibrotic NASH closely resembling clinical features that accompany NASH in humans (*Morrison et al., 2015*; *Liang et al., 2014*). Moreover, these mice show human-like responses to both lipid-lowering and anti-inflammatory therapeutics during the development of metabolic syndrome (*van den Hoek et al., 2014*; *van der Hoorn et al., 2009*; *Li et al., 2018*; *Duivenvoorden et al., 2006*). Here, we show that specific overexpression of FGF21 in the liver, resulting in increased circulating FGF21 levels, activates hepatic signaling associated with FA oxidation and cholesterol removal. In parallel, FGF21 activates thermogenic tissues and reduces adipose tissue inflammation, thereby protecting against adipose tissue dysfunction, hyperglycemia, and hypertriglyceridemia. As a consequence, FGF21 largely limits lipid accumulation in the liver and potently blocks hepatic KC activation and monocyte recruitment, thereby preventing the accumulation of pro-inflammatory macrophages in the liver. In addition, FGF21 reduced the number of pro-fibrotic macrophages in the injured liver, potentially explaining why FGF21 counteracts all features of NASH, including hepatic steatosis, inflammation, and fibrogenesis.

## Results

### Liver-specific FGF21 overexpression increases circulating FGF21 levels and protects against HFCD-induced body fat mass gain

We aimed to elucidate the underlying mechanisms of FGF21-mediated hepatoprotective effects on NASH, by using APOE*3-Leiden.CETP mice fed with an HFCD, a model that induces all stages of

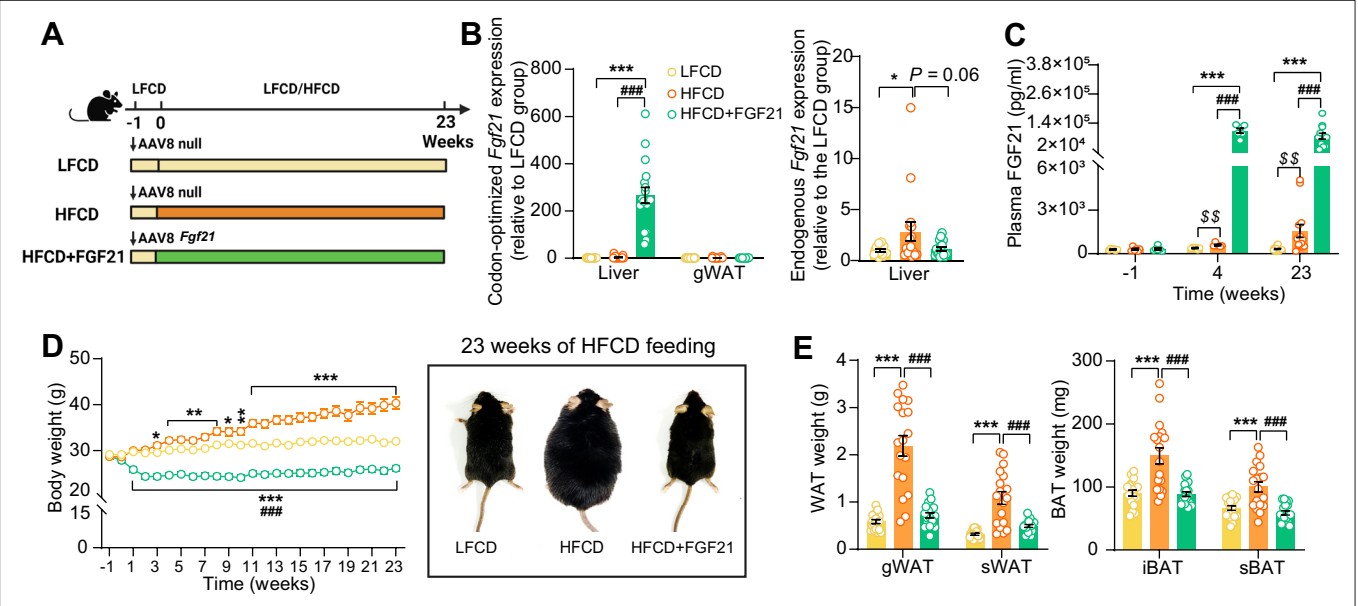

**Figure 1.** Liver-specific FGF21 overexpression increases circulating FGF21 levels and protects against HFCD-induced body fat mass gain. (**A**) Experimental setup. (**B**) At week 23, codon-optimized FGF21 mRNA expression in the liver and gWAT was quantified (n=16 –18), and endogenous *Fgf21* expression in the liver was also measured (n=16–18). (**C**) Plasma FGF21 levels were measured before (at week –1; pooled samples, n=6 per group) and after (at week 4, pooled samples, n=6 per group; week 23, n=12–16 per group) AAV8-*Fgf21* administration. (**D**) Body weight was monitored throughout the experimental period (n=17–18). (**E**) At week 23, brown adipose tissue (BAT) and white adipose tissue (WAT) depots were isolated and weighed (n=18). Data are shown as mean ± SEM. Differences were assessed using one-way ANOVA followed by a Tukey's post test. *p<0.05, **p<0.01, ***p<0.001, compared with the LFCD group. ###p<0.001, compared with the HFCD group. (**C**) Differences between the LFCD and HFCD groups were assessed using Student's t test. $^{\$\$}$p<0.01, compared the LFCD group. AAV8, adeno-associated virus 8; FGF21, fibroblast growth factor 21; gWAT, gonadal WAT; HFCD, high-fat and high-cholesterol diet; iBAT, interscapular BAT; LFCD, low-fat and low-cholesterol diet; sBAT, subscapular BAT; sWAT, subcutaneous WAT.

The online version of this article includes the following source data for figure 1:

**Source data 1.** Liver-specific FGF21 overexpression increases circulating FGF21 levels and protects against HFCD-induced body fat mass gain.

NASH in a human-like fashion and recapitulates the ultrastructural changes observed in NASH patients (*Morrison et al., 2015*; *Liang et al., 2014*). Since the liver is the main contributor to circulating FGF21 (*Fisher and Maratos-Flier, 2016*), we employed an adeno-associated virus 8 (AAV8) vector expressing codon-optimized murine *Fgf21* to induce liver-specific FGF21 overexpression in APOE*3-Leiden. CETP mice. Mice treated with either AAV8-*Fgf21* or AAV8-null as controls were fed with an HFCD for 23 weeks (*Figure 1A*). We confirmed liver-specific FGF21 overexpression by a large increase in codon-optimized *Fgf21* expression in the liver but not in adipose tissue (*Figure 1B*), resulting in high circulating FGF21 levels that persisted throughout the study (*Figure 1C*). In addition, we observed that HFCD feeding increased hepatic endogenous *Fgf21* expression (+184%), which, however, was prevented by AAV8-*Fgf21* administration (*Figure 1B*). Furthermore, by performing a Student's t test between the low-fat low-cholesterol diet (LFCD) and HFCD groups, we did observe that as compared to the LFCD group, HFCD feeding increased plasma FGF21 levels at week 4 (+52%) and week 23 (+383%) (*Figure 1C*). These results are in agreement with previous findings showing that FGF21 is a stress-induced hepatokine whose levels increase in metabolically compromised states, such as obesity (*Zhang et al., 2008*) and NAFLD (*Barb et al., 2019*). HFCD progressively and profoundly increased body weight over the experimental period, accompanied by increased white adipose tissue (WAT) and brown adipose tissue (BAT) weights relative to those of LFCD-fed mice (*Figure 1D,E*). In favorable contrast, FGF21 reduced body weight in the first 3 weeks, after which body weight stabilized and remained lower than that of LFCD- and HFCD-fed mice by the end of the study (−18% and −35%, respectively; *Figure 1D*). Concomitantly, FGF21 decreased weights of gonadal WAT (gWAT; −67%), subcutaneous WAT (sWAT; −55%), interscapular BAT (iBAT; −41%), and subscapular BAT (−41%) to levels comparable to those observed in LFCD-fed mice (*Figure 1E*). These findings thus highlight the potent effects of FGF21 on preventing fat mass gain under NASH-inducing dietary conditions.

## FGF21 protects against HFCD-induced adipose tissue dysfunction

The profound fat mass-lowering effects of liver-derived FGF21 prompted us to examine its role in adipose tissue function. Since we and others have previously shown that FGF21 activates thermogenic adipose tissues (*Schlein et al., 2016*; *Liu et al., 2022*), we first performed histological analyses of BAT and sWAT, the adipose tissue that is most prone to browning (*Zhang et al., 2018*). We observed that FGF21 prevented the HFCD-induced lipid overload in BAT (−66%) and increased uncoupling protein-1 (UCP-1) expression compared with both the LFCD- and HFCD-fed groups (+15% and +26%, respectively) (*Figure 2A*). In sWAT, FGF21 prevented HFCD-induced adipocyte hypertrophy (−41%), and increased the UCP-1 content (+94%) (*Figure 2B*). Among the adipose tissue depots, gWAT is most prone to diet-induced inflammation, and surgical removal of inflamed gWAT attenuates NASH in obese mice (*Mulder et al., 2016*). Similar to sWAT, FGF21 protected against HFCD-induced adipocyte enlargement (−52%) in gWAT and in addition fully prevented the formation of crown-like structures (CLSs; −93%) (*Figure 2C*). In agreement with these findings, FGF21 suppressed the HFCD-induced expression of adhesion G protein-coupled receptor E1 (*Adgre1*; −56%), encoding the macrophage surface marker F4/80, in addition to decreased expression of the pro-inflammatory mediators tumor necrosis factor α (*Tnfa*; −60%), interleukin-1β (*Il1b*; −50%), and monocyte attractant chemokine C–C motif ligand 2 (*Ccl2*; −60%) (*Figure 2D*). Besides, FGF21 tended to upregulate *Klb* (+33%) and *Fgfr1* (+30%) expression compared to HFCD-fed mice (*Figure 2—figure supplement 1*). Moreover, consistent with the critical role of adiponectin in mediating the therapeutic benefits of FGF21 in adipose tissue (*Bao et al., 2018*; *Lin et al., 2013*), FGF21 increased plasma adiponectin levels compared to both LFCD- and HFCD-fed mice (+93% and +133%, respectively; *Figure 2E*). These combined findings thus indicate that FGF21 prevents HFCD-induced adipose tissue dysfunction during NASH development.

## FGF21 alleviates HFCD-induced hyperglycemia and hypertriglyceridemia

We next examined whether FGF21 confers its glucose- and lipid-lowering effects during NASH development. While HFCD induced hyperglycemia as compared to LFCD, FGF21 normalized fasting plasma glucose compared to LFCD, which was accompanied by lower glucose excursion after an intraperitoneal glucose tolerance test (IPGTT) (*Figure 3A,B*). In addition, FGF21 normalized the plasma insulin and Homeostatic Model Assessment for Insulin Resistance index (*Figure 3C*), indicating that FGF21 restores insulin sensitivity to that observed in LFCD-fed mice. FGF21 did not prevent the HFCD-induced increase of plasma total cholesterol (TC) levels (*Figure 3—figure supplement 1A*), nor the distribution of cholesterol over the various lipoproteins (*Figure 3—figure supplement 1B*). Nonetheless, FGF21 strongly and consistently reduced fasting plasma triglyceride (TG) levels throughout the experimental period compared with LFCD- and HFCD-fed mice (−67% and −58%; at week 22), which was specific for very low-density lipoprotein (VLDL) and low-density lipoprotein (LDL) (*Figure 3D*). In addition, an oral lipid tolerance test revealed that FGF21 prevented HFCD-induced lipid intolerance (*Figure 3E*). Taken together, FGF21 prevents the HFCD-induced increase in circulating glucose and reduces circulating TG levels beyond those observed in LFCD-fed mice.

## FGF21 protects against HFCD-induced hepatic steatosis, inflammation, and fibrogenesis

Then, we investigated the effects of FGF21 on liver steatosis, inflammation, and fibrosis. FGF21 not only prevented HFCD-induced liver weight gain (−58%), but even reduced liver weight to a level lower than that of LFCD-fed mice (−40%; *Figure 4A,F*). Moreover, FGF21 abolished the HFCD-induced increase in steatosis, lobular inflammation, and hepatocellular ballooning (*Figure 4B*, *Figure 4—figure supplement 1A,B*). Therefore, FGF21 completely prevented the HFCD-induced large increase in the NAFLD activity score (−74%; *Figure 4C,F*). Furthermore, FGF21 prevented collagen accumulation in the liver as assessed by Picrosirius Red staining (−58%; *Figure 4D,F*). We then measured hepatic concentration of hydroxyproline, a major constituent of collagen and thus a marker of extracellular matrix accumulation. In line with the hepatic collagen content, HFCD feeding increased the hepatic hydroxyproline content, which was prevented by FGF21 (−49%; *Figure 4E*). Taken together, our data demonstrate that FGF21 protects against HFCD-induced hepatosteatosis, steatohepatitis, as well as fibrogenesis.

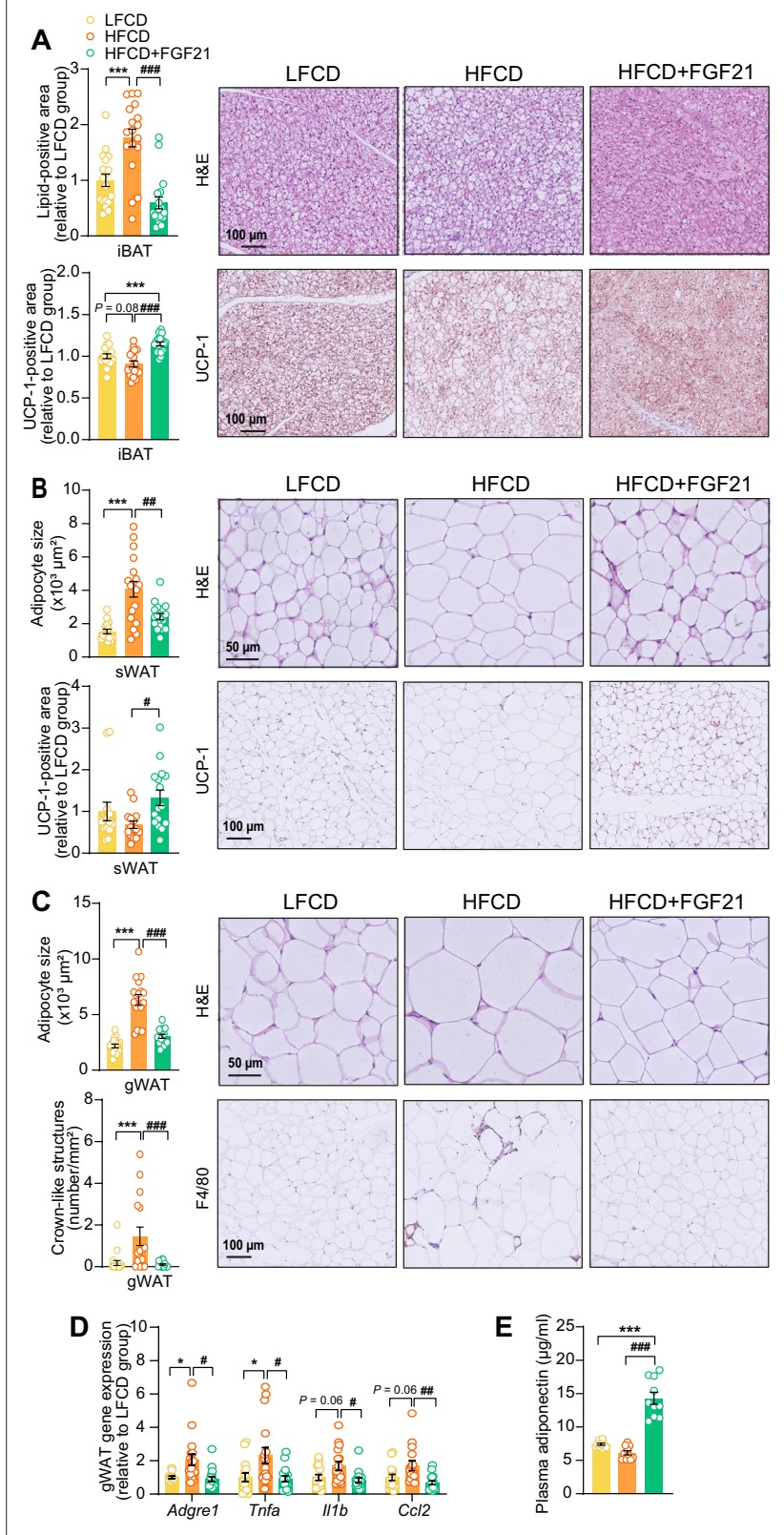

**Figure 2.** Fibroblast growth factor 21 (FGF21) protects against high-fat high-cholesterol diet (HFCD)-induced adipose tissue dysfunction. (**A**) In interscapular brown adipose tissue (iBAT), the lipid content and expression of uncoupling protein-1 (UCP-1) were quantified after hematoxylin-eosin (H&E) staining and UCP-1 immunostaining, respectively. (**B**) In subcutaneous white adipose tissue (sWAT), the adipocyte enlargement was assessed by H&E

*Figure 2 continued on next page*

*Figure 2 continued*

staining, and the tissue browning was evaluated by UCP-1 immunostaining. (**C**) In gonadal white adipose tissue (gWAT), the adipocyte hypertrophy was detected, and the number of crown-like structures (CLSs) was assessed, and (**D**) mRNA expression of pro-inflammatory markers was quantified. (**E**) Plasma adiponectin concentration in fasted blood plasma was measured at week 22. (**A–D**) n=14–18 per group; (**E**) n=10 per group. Differences were assessed using one-way ANOVA followed by a Tukey's post test. *p<0.05, ***p<0.001, compared with the low-fat low-cholesterol diet (LFCD) group. #p<0.05, ##p<0.01, ###p<0.001, compared with the HFCD group. *Adgre1*, adhesion G protein-coupled receptor E1; *Tnfa*, tumor necrosis factor α; *Il1b*, interleukin-1β; *Ccl2*, chemokine C–C motif ligand 2.

The online version of this article includes the following source data and figure supplement(s) for figure 2:

**Source data 1.** Fibroblast growth factor 21 (FGF21) protects against high-fat high-cholesterol diet (HFCD)-induced adipose tissue dysfunction.

**Figure supplement 1.** Liver-specific fibroblast growth factor 21 (FGF21) overexpression tends to upregulate mRNA expression of FGF21 receptor 1 (FGFR1) and co-receptor β-Klotho (KLB) in white adipose tissue (WAT).

**Figure supplement 1—source data 1.** Liver-specific fibroblast growth factor 21 (FGF21) overexpression tends to upregulate mRNA expression of FGF21 receptor 1 (FGFR1) and co-receptor β-Klotho (KLB) in white adipose tissue (WAT).

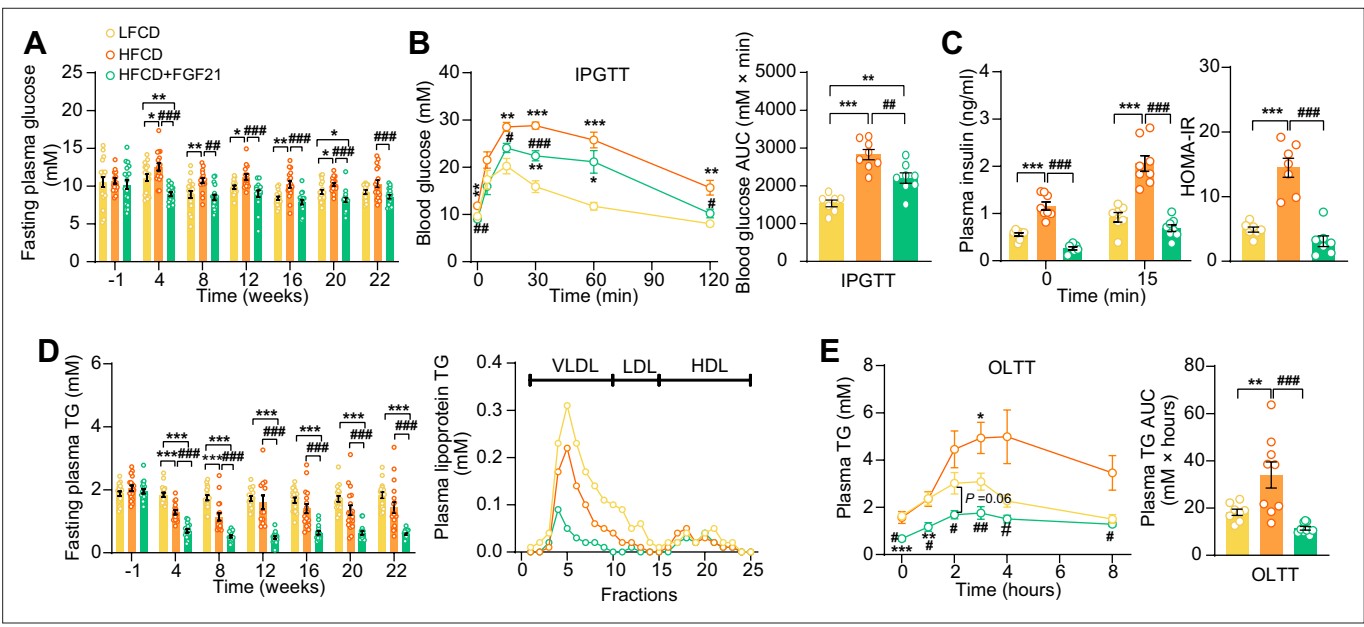

**Figure 3.** Fibroblast growth factor 21 (FGF21) alleviates high-fat high-cholesterol diet (HFCD)-induced hyperglycemia and hypertriglyceridemia. (**A**) Fasting plasma glucose levels were measured during the experimental period. (**B**) At week 16, an intraperitoneal glucose tolerance test (IPGTT) was initiated. (**B**) The area under the curve (AUC) of plasma glucose during the IPGTT and (**C**) plasma insulin concentration in response to the IPGTT was determined at the indicated timepoints. (**C**) Homeostasis model assessment of insulin resistance (HOMA-IR) was determined from fasting glucose and insulin levels. (**D**) Fasting plasma triglyceride (TG) levels were measured throughout the study. The distribution of triglyceride over lipoproteins was determined (pooled samples; n=5 per group) from plasma of week 22. (**E**) At week 20, an oral lipid tolerance test (OLTT) was initiated, and AUC of plasma TG during the OLTT was calculated. (**A and D**) n=14–18 per group; (**B–C**) n=7–8 per group; (**E**) n=6–9 per group. Data are shown as mean ± SEM. Differences were assessed using one-way ANOVA followed by a Tukey's post test. *p<0.05, **p<0.01, ***p<0.001, compared with the low-fat low-cholesterol diet (LFCD) group. #p<0.05, ##p<0.01, ###p<0.001, compared with the HFCD group.

The online version of this article includes the following source data and figure supplement(s) for figure 3:

**Source data 1.** Fibroblast growth factor 21 (FGF21) alleviates high-fat high-cholesterol diet (HFCD)-induced hyperglycemia and hypertriglyceridemia.

**Figure supplement 1.** High-fat high-cholesterol diet (HFCD) increases fasting cholesterol levels.

**Figure supplement 1—source data 1.** High-fat high-cholesterol diet (HFCD) increases fasting cholesterol levels.

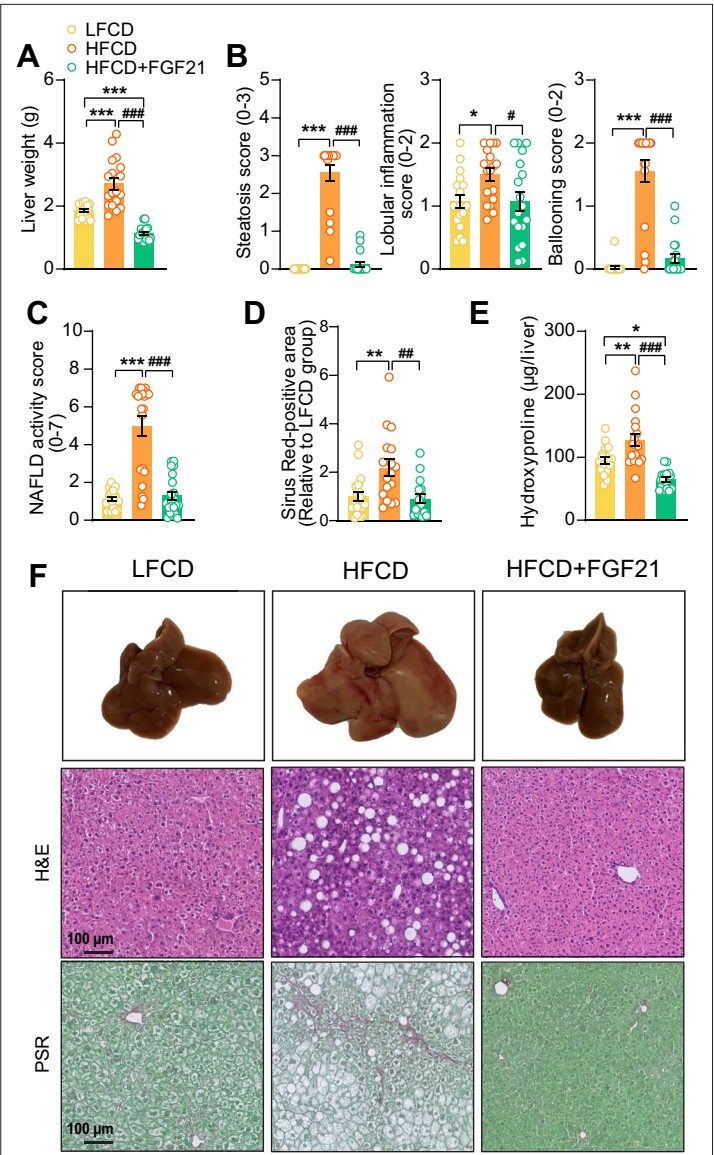

**Figure 4.** Fibroblast growth factor 21 (FGF21) protects against high-fat high-cholesterol diet (HFCD)-induced hepatic steatosis, inflammation, and fibrosis. (**A**) At week 23, liver weight was determined, and (**B**) scoring of histological features of steatosis, lobular inflammation, and ballooning as well as (**C**) nonalcoholic fatty liver disease (NAFLD) activity was evaluated by hematoxylin-eosin (H&E) staining. (**D**) Liver fibrosis was assessed by Picrosirius Red (PSR) staining, and (**E**) hepatic hydroxyproline levels were determined. (**F**) Representative macroscopic, H&E, and PSR pictures are shown. Data are shown as mean ± SEM (n=16–18 per group). Differences were assessed using one-way ANOVA followed by a Tukey's post test. *p<0.05; **p<0.01, ***p<0.001, compared with the low-fat low-cholesterol diet (LFCD) group. ##p<0.01; ###p<0.001, compared with the HFCD group.

The online version of this article includes the following source data and figure supplement(s) for figure 4:

**Source data 1.** Fibroblast growth factor 21 (FGF21) protects against high-fat high-cholesterol diet (HFCD)-induced hepatic steatosis, inflammation, and fibrosis.

**Figure supplement 1.** Fibroblast growth factor 21 (FGF21) abolishes high-fat high-cholesterol diet (HFCD)-induced increase of hepatic lipid-positive area and the number of inflammatory foci.

**Figure supplement 1—source data 1.** Fibroblast growth factor 21 (FGF21) abolishes high-fat high cholesterol diet (HFCD)-induced increase of hepatic lipid-positive area and the number of inflammatory foci.

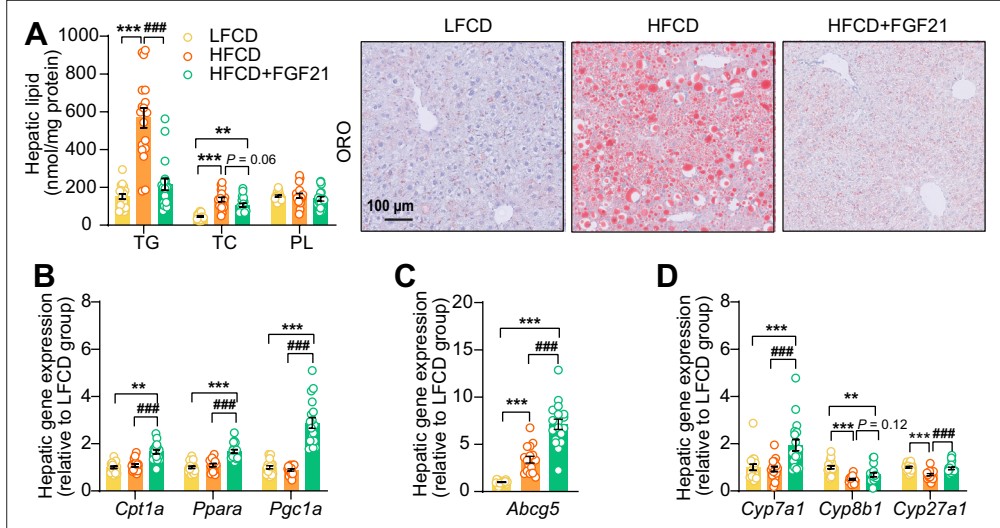

**Figure 5.** Fibroblast growth factor 21 (FGF21) abolishes liver lipotoxicity, accompanied by activation of hepatic signaling involved in fatty acid (FA) oxidation and cholesterol removal. (**A**) Triglyceride (TG), total cholesterol (TC), and phospholipid (PL) levels were determined in the liver (n=18 per group), and representative Oil Red O (ORO) pictures are shown. (**B**) The relative mRNA expression of genes involved in fatty acid oxidation and (**C and D**) cholesterol removal (n=15–18 per group) were determined in the liver. Data are shown as mean ± SEM. Differences were assessed using one-way ANOVA followed by a Tukey's post test. **p<0.01, ***p<0.001, compared with the low-fat low-cholesterol diet (LFCD) group. ###p<0.001, compared with the high-fat high-cholesterol diet (HFCD) group. *Abcg5*, ATP-binding cassette transporter G member 5; *Cpt1a*, carnitine palmitoyl transferase 1α; *Cyp7a1*, cholesterol 7α-hydroxylase; *Cyp8b1*, sterol 12α-hydroxylase; *Cyp27a1*, sterol 27-hydroxylase; *Pgc1a*, peroxisome proliferator-activated receptor gamma coactivator 1α; *Ppara*, peroxisome proliferator-activated receptor α.

The online version of this article includes the following source data and figure supplement(s) for figure 5:

**Source data 1.** Fibroblast growth factor 21 (FGF21) abolishes liver lipotoxicity, accompanied by activation of hepatic signaling involved in fatty acid (FA) oxidation and cholesterol removal.

**Figure supplement 1.** Liver-specific fibroblast growth factor 21 (FGF21) overexpression upregulates hepatic mRNA expression of FGF21 receptors (FGFRs) and co-receptor β-Klotho (KLB).

**Figure supplement 1—source data 1.** Liver-specific fibroblast growth factor 21 (FGF21) overexpression upregulates hepatic mRNA expression of FGF21 receptors (FGFRs) and co-receptor β-Klotho (KLB).

**Figure supplement 2.** Fibroblast growth factor 21 (FGF21) increases apolipoprotein B mRNA (*Apob*) expression in the liver.

**Figure supplement 2—source data 1.** Fibroblast growth factor 21 (FGF21) increases apolipoprotein B mRNA (*Apob*) expression in the liver.

## FGF21 abolishes liver lipotoxicity, accompanied by activation of hepatic signaling involved in FA oxidation and cholesterol removal

In the context of NASH, pro-inflammatory responses and fibrogenesis occur when hepatocytes are injured by lipotoxicity (*Neuschwander-Tetri, 2010*; *Machado and Diehl, 2016*). Indeed, 23 weeks of HFCD feeding promoted aberrant accumulation of TG as well as TC in the liver (*Figure 5A*). In agreement with the data presented in *Figure 4*, FGF21 abrogated the HFCD-induced increase in hepatic TG levels (−62%) and tended to decrease hepatic TC levels (−22%), resulting in smaller lipid droplets (*Figure 5A*). In addition to reduced lipid overflow from WAT, we reasoned that FGF21 may also directly act on the liver to prevent HFCD-induced liver lipotoxicity. In agreement, compared to both LFCD- and HFCD-fed mice, FGF21 profoundly upregulated the expression of *Klb* (+150% and +223%), *Fgfr1* (+57% and +79%), *Fgfr2* (+97% and +77%), and *Fgfr4* (+53% and +67%) (*Figure 5—figure supplement 1*). We next quantified the hepatic expression of key genes involved in FA and cholesterol handling. FGF21 did not attenuate the HFCD-induced increased expression of FA translocase cluster of differentiation 36 (C*d36*) (*Figure 5—figure supplement 2A*). In favorable

contrast, compared to both LFCD- and HFCD-fed mice, FGF21 did increase the expression of carnitine palmitoyl transferase 1α (*Cpt1a*, +66% and +53%), peroxisome proliferator-activated receptor α (*Ppara*, +67% and +53%) and peroxisome proliferator-activated receptor γ coactivator 1α (*Pgc1a*; +188% and +225%), all of those genes being key players involved in FA oxidation (*Figure 5B*). Moreover, compared to LFCD- and HFCD-fed mice, FGF21 increased the expression of apolipoprotein B (*Apob*, +26% and +38%), which is involved in VLDL secretion (*Figure 5—figure supplement 2B*). Furthermore, FGF21 upregulated the expression of ATP-binding cassette transporter G member 5 (*Abcg5*; sevenfold and twofold), crucial for biliary secretion of neutral sterols (*Figure 5C*), increased the expression of cholesterol 7α-hydroxylase (*Cyp7a1*; +94% and +109%), a key gene involved in the classic bile acid synthesis pathway (*Figure 5D*), and restored the expression of sterol 27-hydroxylase (+38%), involved in the alternative bile acid pathway (*Figure 5D*). Considering that bile acid synthesis is a major pathway for hepatic cholesterol disposal (*Tu et al., 2000*), FGF21 likely regulates bile acid metabolism to prevent HFCD-induced cholesterol accumulation in the liver. Collectively, our data indicate that FGF21 increases the hepatic expression of key genes involved in β-oxidation and cholesterol removal, which together with reduced lipid overload from WAT may explain FGF21-induced alleviation of liver lipotoxicity under NASH-inducing dietary conditions.

## FGF21 prevents activation of various KC subsets

Then, we performed an in-depth phenotyping of hepatic immune cells using spectral flow cytometry. For this, we developed a panel that identifies most major immune cell subsets (for gating strategy see *Figure 6—figure supplement 1A*). As compared to LFCD, HFCD tended to reduce total CD45[+] leukocytes, which were increased by FGF21 (*Figure 6—figure supplement 1B*). Combining conventional gating and dimension reduction analysis through uniform manifold approximation and projection allowed to identify FGF21-induced changes in cell subset abundance (*Figure 6A*). FGF21 prevented HFCD-induced loss of eosinophils, neutrophils and B cells, and increased numbers of dendritic cells and T cells compared with those observed in both LFCD- and HFCD-fed mice (*Figure 6—figure supplement 1B*). More importantly, FGF21 increased the number of total KCs compared with that of both LFCD- and HFCD-fed mice (+63% and +156; *Figure 6—figure supplement 1B*), attenuated HFCD-induced monocyte recruitment (−18%), and tended to repress the HFCD-induced increase in hepatic MoDMacs (−42%; *Figure 6—figure supplement 1B*).

During the development of NASH, MoDMacs can gradually seed in KC pool by acquiring ResKCs identity and replacing the dying ResKCs (*Tran et al., 2020*). These recruited MoKCs can have both detrimental and supportive roles, contributing to increase in pathology during fibrosis onset, but hastening recovery when the damage-evoking agent is attenuated/removed (*Seidman et al., 2020*). In light of this, we assessed the abundance and phenotype of ResKCs and monocyte-derived KCs (MoKCs). We observed that FGF21 completely abolished the HFCD-induced reduction of the number of ResKCs (+319%) and potently protected against HFCD-induced ResKC activation as shown by decreased proportion of CD11c[+] ResKCs (−53%; *Figure 6B*). FGF21 also completely abolished the HFCD-induced upregulation of CD36 in ResKCs, to levels that are even lower than those in LFCD-fed mice (−88% vs. LFCD; −94% vs. HFCD; *Figure 6B*). In addition, FGF21 increased the number of MoKCs compared with that of both LFCD- and HFCD-fed mice (+92% and +123%), and prevented the HFCD-induced increase in the abundance of CD11c[+] MoKCs (−42%) (*Figure 6C*). Strikingly, compared to both LFCD- and HFCD-fed mice, FGF21 downregulated CD9 (−32% and −49%) and CD36 (−98% and −100%) in MoKCs (*Figure 6C*). Furthermore, FGF21 profoundly repressed HFCD-induced upregulation of hepatic *Tnfa* (−37%), *Il1b* (−41%), and *Ccl2* (−54%) expression to levels comparable to those in LFCD-fed mice (*Figure 6D*), which is in line with the observation that FGF21 prevents KC activation. Given that CD36[hi] ResKCs and CD36[hi]/ CD9[hi] MoKCs are involved in the formation of hepatic CLSs (*Tran et al., 2020*; *Seidman et al., 2020*; *Blériot et al., 2021*; *Daemen et al., 2021*), we next assessed CLSs and observed that FGF21 completely prevented the HFCD-induced formation of CLSs in the liver (−93%; *Figure 6D*). These data demonstrate that FGF21 inhibits the activation of ResKCs and MoKCs and prevents the accumulation of CD36[hi] ResKCs and CD36[hi]/CD9[hi] MoKCs under dietary conditions that result in NASH, which likely contribute to the beneficial effects of FGF21 on hepatic inflammation and fibrosis.

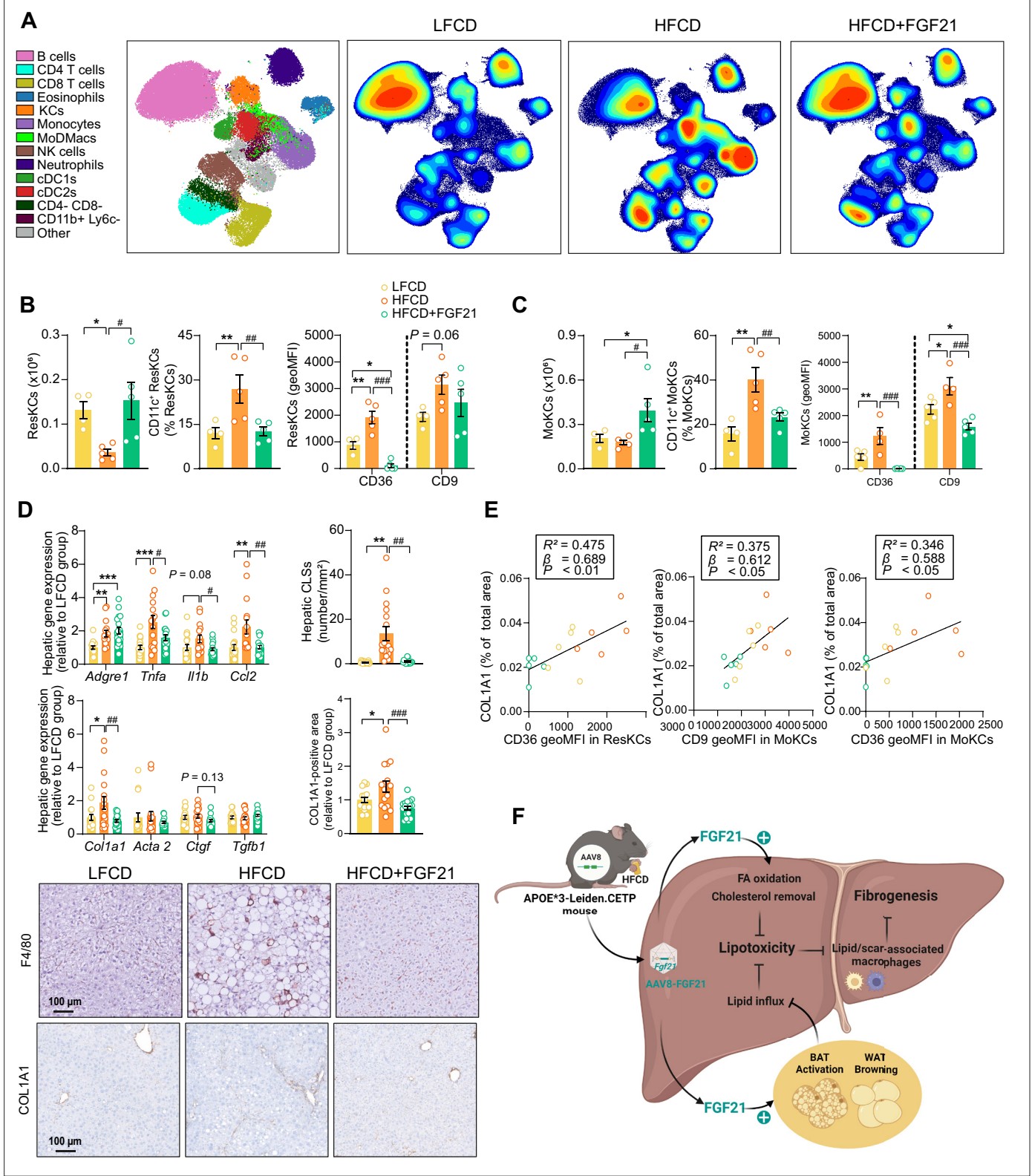

**Figure 6.** Fibroblast growth factor 21 (FGF21) modulates hepatic macrophage pool and protects against COL1A1 accumulation, as predicted by the reduction of CD36hi Kupffer cells (KCs) and CD9hi KCs. (**A**) Uniform manifold approximation and projection for dimension reduction (UMAP) of immune cell subsets from livers after 23 weeks of intervention. (**B**) The number of resident KCs (ResKCs), the proportion of CD11c+ ResKCs, and the expression of CD36 and CD9 in ResKCs were quantified. (**C**) The amount of monocyte-derived KCs (MoKCs) was assessed, the percentage of CD11c+ MoKCs

*Figure 6 continued on next page*

*Figure 6 continued*

was determined, the CD36 and CD9 expression levels in MoKCs were quantified. (**D**) Hepatic inflammation was evaluated by pro-inflammatory gene expression and the formation of crown-like structures (CLSs) within the liver. The mRNA expression of liver fibrogenesis markers was quantified, and the protein expression of collagen type 1α 1 (COL1A1) was determined. (**E**) The expression of CD36 in ResKCs, and the expression of CD9 and CD36 in MoKCs were plotted against COL1A1-positive area in the liver. (**F**) Mechanistic model. Data are shown as mean ± SEM (**A–B and E**, n=4–5 per group; **D**, n=16–18 per group). Linear regression analyses were performed. Differences were assessed using one-way ANOVA followed by a Fisher's LSD test. *p<0.05, **p<0.01, ***p<0.001, compared with the low-fat low-cholesterol diet (LFCD) group. #p<0.05, ##p<0.01, ###p<0.001, compared with the high-fat high-cholesterol diet (HFCD) group. *Acta2*, actin α2; *Ctgf*, connective tissue growth factor; FA, fatty acid; *Tgfb1*, transforming growth factor-β.

The online version of this article includes the following source data and figure supplement(s) for figure 6:

**Source data 1.** Fibroblast growth factor 21 (FGF21) modulates hepatic macrophage pool and protects against COL1A1 accumulation, as predicted by the reduction of CD36[hi] Kupffer cells (KCs) and CD9[hi] KCs.

**Figure supplement 1.** Fibroblast growth factor 21 (FGF21) modulates the hepatic immune cell pool.

**Figure supplement 1—source data 1.** Fibroblast growth factor 21 (FGF21) modulates the hepatic immune cell pool.

**Figure supplement 2.** CD36[hi] resident Kupffer cells (ResKCs) as well as CD36[hi]/CD9[hi] monocyte-derived KCs (MoKCs) positively correlate with nonalcoholic fatty liver disease (NAFLD) activity score and liver fibrosis.

**Figure supplement 2—source data 1.** CD36[hi] resident Kupffer cells (ResKCs) as well as CD36[hi]/CD9[hi] monocyte-derived KCs (MoKCs) positively correlate with nonalcoholic fatty liver disease (NAFLD) activity score and liver fibrosis.

## FGF21 protects against COL1A1 accumulation, as predicted by the reduction of CD36[hi] KCs and CD9[hi] KCs

To further evaluate whether FGF21-induced reductions of lipid-associated macrophages (i.e., CD36[hi] ResKCs and CD36[hi] MoKCs) (*Blériot et al., 2021*) and scar-associated macrophages (i.e., CD9[hi] MoKCs) (*Ramachandran et al., 2019*) are implicated in fibrogenesis, we performed multiple univariate regression analyses. These revealed that both NAFLD activity and liver fibrosis were associated with both CD36[hi] ResKCs, CD36[hi] MoKCs, and CD9[hi] MoKCs (*Figure 6—figure supplement 2A-D*), indicating that FGF21 likely improves liver fibrosis by reducing these lipid- and scar-associated macrophages. To further understand the underlying mechanisms by which FGF21 prevents liver fibrosis, we measured hepatic expression of key genes involved in fibrogenesis (*Figure 6D*). FGF21 tended to decrease the expression of connective tissue growth factor (*Ctgf*; −27%), a major fibrogenic factor, and normalized the HFCD-induced increased expression of its downstream target collagen type Iα 1 (*Col1a1*; −61%; *Figure 6D*). This finding was confirmed by immunohistochemistry, revealing that FGF21 reduced hepatic COL1A1 accumulation (−46%; *Figure 6D*). Furthermore, univariate regression analysis revealed that COL1A1 expression is predicted by CD36[hi] ResKCs, CD36[hi] MoKCs, and CD9[hi] MoKCs (*Figure 6E*, *Figure 6—figure supplement 2E*). Taken together, these data indicate that FGF21 reduces lipid- and scar-associated macrophages to inhibit COL1A1 synthesis and prevent fibrogenesis.

## Discussion

Several FGF21 analogues are currently being evaluated in clinical trials for the treatment of NASH (*Sanyal et al., 2019*; *Harrison et al., 2021*). While the protective effect of pharmacological intervention with long-acting FGF21 on human liver steatosis has been uncovered (*Sanyal et al., 2019*; *Harrison et al., 2021*; *Aggarwal et al., 2022*), mechanisms underlying attenuated steatosis as well all the anti-inflammatory and anti-fibrotic effects of FGF21 on NASH are still largely unexplored. Therefore, we set out to elucidate mechanisms by which FGF21 beneficially modulates these various aspects of NASH in HFCD-fed APOE*3-Leiden.CETP mice, a well-established model for diet-induced NASH (*Morrison et al., 2015*; *Liang et al., 2014*). Based on our findings, we propose that FGF21 attenuates liver lipotoxicity via endocrine signaling to adipose tissue to induce thermogenesis, thereby preventing adipose tissue dysfunction to reduce lipid overflow to the liver, as well as autocrine signaling to the liver to increase FA oxidation and cholesterol removal. In addition, FGF21 prevents KC activation, monocyte recruitment, and the formation of lipid- and scar-associated macrophages, thereby likely inhibiting collagen accumulation and alleviating liver fibrogenesis.

Hepatic lipotoxicity is one of the major risk factors determining the progression of liver steatosis into NASH, as shown in multiple clinical studies with obese patients (*Bril et al., 2017*; *Armstrong et al.,*

2016; *Ratziu et al., 2021*). By feeding APOE*3-Leiden.CETP mice a diet rich in fat and cholesterol, we mimicked a situation in which a positive energy balance induces many aspects of the metabolic syndrome, including insulin resistance, obesity with increased fat accumulation, and hepatic lipotoxicity indicated by hepatomegaly with aberrant accumulation of TG as well as TC. Hepatic lipotoxicity likely results from lipid overflow from insulin-resistant adipose tissue toward the liver in combination with hepatic insulin resistance that prevents insulin-stimulated outflow of lipids (*Zarei et al., 2020*). Within this dietary context, we applied a single administration of an AAV8 vector encoding codon-optimized FGF21, which resulted in liver-specific FGF21 overexpression. Since the codon-optimized FGF21 mitigates the poor pharmacokinetic properties of native FGF21, including its short plasma half-life (0.5–2 hr) by reducing proteolytic degradation (*Zarei et al., 2020*), an elevated level of circulating FGF21 was reached throughout the dietary intervention period. By this strategy, we mimicked the situation in which circulating FGF21 predominantly derives from the liver (*Nishimura et al., 2000*). Indeed, circulating FGF21 correlates well with the hepatic expression of FGF21 (*Markan et al., 2014*). Interestingly, hepatic expression of FGF21 fully prevented the diet-induced increase in liver weight, liver lipids (i.e., TG and TC), and steatosis score.

These lipotoxicity-protective effects of FGF21 can partially be explained by endocrine effects of liver-derived FGF21 on adipose tissue, which besides the liver has high expression of KLB, the co-receptor of the FGFR (*Fisher and Maratos-Flier, 2016*; *Geng et al., 2020*). Indeed, FGF21 fully prevented the HFCD-induced increase in weights of WAT and BAT, with decreased lipid accumulation in these adipose tissue depots as well as induction of BAT activation and WAT browning. These data imply that FGF21 induces thermogenesis which increases energy expenditure, consistent with the thermogenic responses observed for recombinant FGF21 in C57BL/6 mice fed with an obesogenic diet (*Schlein et al., 2016*). Likewise, by using APOE*3-Leiden.CETP mice, we previously reported that FGF21 treatment highly increased energy expenditure without affecting food intake (*Liu et al., 2022*). Activation of thermogenic tissues by classical β-adrenergic receptor largely increases the uptake of circulating lipoprotein-derived FAs by BAT and beige WAT (*Berbée et al., 2015*), which we recently also demonstrated for recombinant FGF21 (*Liu et al., 2022*). This can thus at least partly explain the marked TG-lowering effect of FGF21 observed in the current study. Thermogenic activation also increases the uptake and combustion of glucose, although the glucose-lowering and insulin-sensitizing effects of FGF21 can also be explained by attenuated WAT inflammation in combination with increased adiponectin expression as well as improved liver insulin sensitivity (*Liu et al., 2022*; *Lin et al., 2013*; *Yang et al., 2018*).

Besides endocrine FGF21 signaling in adipose tissue, liver lipotoxicity is likely further prevented by autocrine FGF21 signaling. Indeed, we showed that liver-specific FGF21 overexpression increased hepatic expression of genes involved in FA oxidation (*Cpt1a*, *Ppara*, *Pgc1a*), biliary cholesterol secretion (*Abcg5*), bile acids synthesis (*Cyp7a1*), and VLDL production (*Apob*). Of note, these observations are in line with previous reports showing increased FA oxidation (*Fisher et al., 2014*) and upregulated *Abcg5* (*Keinicke et al., 2020*), *Cyp7a1* (*Keinicke et al., 2020*; *Zhang et al., 2017*), and *Apob* (*Liu et al., 2022*) in the liver upon FGF21 treatment. Altogether, the marked protective effects of FGF21 on HFCD-induced hepatic lipotoxicity likely results from combined endocrine and autocrine signaling, leading to reduced lipid influx from adipose tissue to the liver coupled to the activation of hepatic FA oxidation and cholesterol elimination pathways. Our observations may likely explain the recent clinical findings that treatment with FGF21 analogues in patients with NASH not only reduced hepatic steatosis (*Sanyal et al., 2019*; *Harrison et al., 2021*) but also increased hepatic bile acid synthesis and further promoted cholesterol removal, lowering the risk for further hepatic lipotoxicity (*Luo et al., 2022*).

While NASH is initiated by hepatic lipotoxicity, NASH progression is mainly driven by impaired KC homeostasis and subsequent liver inflammation (*Cai et al., 2019*). Therefore, we investigated in depth the inflammatory response in the liver through a combination of immunohistochemistry, flow cytometry, and gene expression analyses. HFCD feeding induced an array of inflammatory effects, including increased lobular inflammation, hepatocyte ballooning and NAFLD activity scores, as well as increased inflammatory foci and CLSs, accompanied by a reduction in ResKCs with a relative increase in CD11c+ ResKCs, and an increase in MoDMacs and CD11c+ MoKCs. These observations are likely explained by lipotoxicity-related damage to ResKCs, and release of TNFα, IL-1β, and MCP-1 (*Ccl2*), both activating various downstream pro-inflammatory mediators and promoting monocyte recruitment to remodel

the KC pool (*Tran et al., 2020*; *Remmerie et al., 2020*) and further exacerbating hepatic inflammation (*Tran et al., 2020*; *Blériot et al., 2021*; *Cai et al., 2019*; *Schwabe et al., 2020*; *Yu et al., 2019*). Importantly, FGF21 prevented most of these HFCD-induced inflammatory responses, as it normalized lobular inflammation, hepatocyte ballooning and NAFLD activity scores and CLSs, and reduced pro-inflammatory activation of various KC subsets.

Fibrosis has been identified as the most important predictor of prognosis in NAFLD patients, and therefore a main target in experimental pharmacological approaches (*Heyens et al., 2021*). HFCD feeding during 23 weeks induced early signs of fibrosis, as evident from an increased *Col1a1* expression and COL1A1 content, accompanied by an increased content of the hydroxyproline. Importantly, FGF21 blocked liver fibrogenesis, and decreased the hydroxyproline content. These alterations were accompanied with reductions in lipid-associated macrophages (i.e., CD36hi ResKCs/MoKCs) (*Blériot et al., 2021*) and scar-associated macrophages (i.e., CD9hi MoKCs) (*Ramachandran et al., 2019*). In fact, when analyzing the mouse groups together, CD36hi ResKCs/MoKCs and CD9hi MoKCs positively correlated with liver fibrosis as reflected by hydroxyproline content and COL1A-positive area, suggesting that these lipid- and scar-associated macrophages are involved in fibrogenesis in our model. Indeed, high numbers of CD9hi macrophages have been found in fibrotic regions of the liver (*Seidman et al., 2020*; *Daemen et al., 2021*; *Ramachandran et al., 2019*; *Remmerie et al., 2020*), and these cells are able to prime quiescent primary murine hepatic stellate cells to upregulate the expression of fibrillar collagen through CTGF (*Ramachandran et al., 2019*), thereby promoting and exacerbating liver fibrosis. Therefore, we speculate that FGF21 protects against early liver fibrosis likely through preventing the accumulation of CD36hi/CD9hi KCs, thereby inhibiting activation of hepatic stellate cells to produce collagen.

This study is not without limitations. In this work, we used a gene therapy approach to examine the effects of liver-derived FGF21 on NASH based on the use of a single injection of an AAV8 vector encoding codon-optimized murine FGF21. Although AAV8 is hepatocyte trophic, we have not excluded potential contribution of other hepatic cells to total FGF21 expression. Also, while AAV8-*Fgf21* was non-toxic, sustained supra-pharmacological plasma levels of FGF21 were achieved, which do not necessarily reflect effects of current pharmacological strategies with long-acting FGF21. Interestingly, AAV-mediated gene therapy has already been tested in the clinic for life-threatening diseases such as hemophilia B, and has demonstrated stable expression of factor IX following AAV-mediated delivery (*Nathwani et al., 2018*). Therefore, it is reasonable to speculate that liver-targeted gene therapy as an approach to induce stable overexpression of FGF21 may ultimately have potential to reach to the clinic.

In conclusion, hepatic overexpression of FGF21 in APOE*3-Leiden.CETP mice limits diet-induced hepatic lipotoxicity, inflammation, and fibrogenesis. Through a combination of endocrine and autocrine signaling, FGF21 reduces hepatic lipid influx and accumulation, respectively. This results in reduced macrophage activation and monocyte recruitment with less presence of lipid- and scar-associated macrophages, limiting activation of hepatic stellate cells to produce collagen (for graphic summary, see *Figure 6F*). As such, our studies provide a mechanistic explanation for the hepatoprotective effects of FGF21 analogues in recent clinical trials including reduction in steatosis (*Sanyal et al., 2019*; *Harrison et al., 2021*; *Luo et al., 2022*) as well as the fibrotic marker N-terminal type III collagen pro-peptide (*Sanyal et al., 2019*; *Harrison et al., 2021*), and further highlight the potential of FGF21 for clinical implementation as a therapeutic in the treatment of advanced NASH.

## Materials and methods

Please see Appendix 1 for a detailed description of all experimental procedures.

### Animals and treatments

Male APOE*3-Leiden.CETP mice (on a C57BL/6J background) were generated as previously described (*Westerterp et al., 2006*). Mice at the age of 10–12 weeks were group-housed (2–4 mice per cage) under standard conditions (22°C, 12/12 hr light/dark cycle) with ad libitum access to water and an LFCD (Standard Rodent Diet 801203, Special Diets Services, UK), unless indicated otherwise. Then, based on body weight and 4 hr (9.00–13.00) fasted plasma glucose, TG and TC levels, these mice were randomized into three treatment groups (n=18 per group), after which they received either

AAV8-*Fgf21*, a liver-tropic AAV8 capsid vector expressing codon-optimized murine *Fgf21* under the control of a liver-specific apolipoprotein E (*Apoe*)/alpha-1-antitrypsin (*Aat*) promoter (HFCD+FGF21 group; $2\times10^{10}$ genome copies per mouse), or with the same genome copy number of AAV8-null (HFCD and LFCD groups) via a single intravenous injection. After 1 week of recovery, mice in the HFCD+FGF21 and HFCD groups were switched to an HFCD (60% fat and 1% cholesterol; C1090-60, Altromin, Germany) and maintained on the diet for 23 weeks, at which APOE*3-Leiden.CETP mice have developed both steatosis, hepatic inflammation and early fibrosis (*Morrison et al., 2015*; *Hui et al., 2018*). An IPGTT (n=8 per group) and an oral lipid tolerance test (n=10 per group) were performed at week 16 and week 20, respectively. Flow cytometry (n=5 per group) was conducted at week 23.

## Statistics

Comparisons among three groups were analyzed using one-way ANOVA followed by a Tukey's post test, unless indicated otherwise. Data are presented as mean ± SEM, and a p-value of less than 0.05 was considered statistically significant. All statistical analyses were performed with GraphPad Prism 9.01 for Windows (GraphPad Software Inc, California, CA, USA).

## Study approval

All animal experiments were carried out according to the Institute for Laboratory Animal Research Guide for the Care and Use of Laboratory Animals, and were approved by the National Committee for Animal Experiments (Protocol No. AVD1160020173305) and by the Ethics Committee on Animal Care and Experimentation of the Leiden University Medical Center (Protocol No. PE.18.034.041).

## Acknowledgements

This work was supported by the Dutch Diabetes Research Foundation (2015.81.1808 to MRB); the Netherlands Organisation for Scientific Research-NWO (VENI grant 91617027 to YW); Chinese Scholarship Council grants (CSC 201606010321 to EZ); the Novo Nordisk Foundation (NNF18OC0032394 to MS); and the Netherlands Cardiovascular Research Initiative: an initiative with support of the Dutch Heart Foundation (CVON-GENIUS-2 to PCNR). The authors also thank TCM Streefland, ACM Pronk, RA Lalai, and HCM Sips from Department of Medicine, the Division of Endocrinology, Leiden University Medical Center for technical assistance.

## Additional information

### Competing interests

Anne-Christine Andreasson, Andrew Park, Stephanie Oldham, Martin Uhrbom, Ingela Ahlstedt, Yasuhiro Ikeda, Kristina Wallenius, Xiao-Rong Peng: employee of AstraZeneca. The other authors declare that no competing interests exist.

### Funding

| Funder | Grant reference number | Author |
| --- | --- | --- |
| Diabetes Fonds | 2015.81.1808 | Mariëtte R Boon |
| Netherlands Organisation for Scientific Research | VENI grant 91617027 | Yanan Wang |
| Chinese Scholarship Council | CSC 201606010321 | Enchen Zhou |
| Novo Nordisk Foundation | NNF18OC0032394 | Milena Schönke |
| Hartstichting | The Netherlands Cardiovascular Research Initiative CVON-GENIUS-2 | Patrick CN Rensen |

| Funder | Grant reference number | Author |
|--------|----------------------|--------|

The funders had no role in study design, data collection and interpretation, or the decision to submit the work for publication.

## Author contributions

Cong Liu, Conceptualization, Data curation, Formal analysis, Investigation, Writing - original draft; Milena Schönke, Conceptualization, Supervision, Validation, Investigation, Methodology, Writing - review and editing; Borah Spoorenberg, Formal analysis, Investigation, Writing - review and editing; Joost M Lambooij, Hendrik JP van der Zande, Data curation, Formal analysis, Methodology, Writing - review and editing; Enchen Zhou, Yasuhiro Ikeda, Investigation, Methodology, Writing - review and editing; Maarten E Tushuizen, Stephanie Oldham, Investigation, Writing - review and editing; Anne-Christine Andreasson, Martin Uhrbom, Ingela Ahlstedt, Formal analysis, Writing - review and editing; Andrew Park, Methodology, Writing - review and editing; Kristina Wallenius, Formal analysis, Validation, Investigation, Writing - review and editing; Xiao-Rong Peng, Conceptualization, Validation, Investigation, Methodology, Writing - review and editing; Bruno Guigas, Conceptualization, Data curation, Formal analysis, Validation, Investigation, Methodology, Writing - review and editing; Mariëtte R Boon, Funding acquisition, Investigation, Writing - review and editing; Yanan Wang, Conceptualization, Data curation, Supervision, Funding acquisition, Validation, Investigation, Methodology, Writing - review and editing; Patrick CN Rensen, Conceptualization, Data curation, Supervision, Funding acquisition, Validation, Investigation, Methodology, Project administration, Writing - review and editing

## Author ORCIDs

Cong Liu ⓘ http://orcid.org/0000-0002-2852-8953
Enchen Zhou ⓘ http://orcid.org/0000-0002-3739-4934
Anne-Christine Andreasson ⓘ http://orcid.org/0000-0002-8323-0658
Kristina Wallenius ⓘ http://orcid.org/0000-0002-3231-2733
Patrick CN Rensen ⓘ http://orcid.org/0000-0002-8455-4988

## Ethics

All animal experiments were carried out according to the Institute for Laboratory Animal Research Guide for the Care and Use of Laboratory Animals, and were approved by the National Committee for Animal Experiments (Protocol No. AVD1160020173305) and by the Ethics Committee on Animal Care and Experimentation of the Leiden University Medical Center (Protocol No. PE.18.034.041).

## Decision letter and Author response

Decision letter https://doi.org/10.7554/eLife.83075.sa1
Author response https://doi.org/10.7554/eLife.83075.sa2

# Additional files

## Supplementary files

• MDAR checklist

## Data availability

All data generated or analysed during this study are included in the manuscript and supporting files.

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

## Appendix 1

### Materials and methods

#### Generation of recombinant AAV vectors

AD-293 cells (Agilent, Santa Clara, CA, USA) were plated in a five layered chamber in Gibco DMEM supplemented with 10% Gibco FBS and 1% Gibco penicillin-streptomycin (Thermo Fisher Scientific, Waltham, MA, USA). When these cells reached at 60–85% confluency under the microscope, they were transfected by polyethylenimine (Polyscience, Torrance, CA, USA) with triple plasmids, including pHelper containing adenoviral E2A and E4 genes, pRep2Cap8 encoding AAV2 Rep proteins and AAV8 serotype capsid, and either pAAV-apolipoprotein E (*Apoe*)/alpha-1-antitrypsin (*Aat*) promoter-driven codon-optimized murine *Fgf21* or pAAV-*Apoe*/no plasmid, in a ratio of 2:1.4:1, respectively. After 72 hr of the post-transfection, cells were harvested and lysed via three freeze-thaw cycles followed by 1 hr of benzonase treatment at 37°C. Supernatants were then further purified using iodixanol gradient-based ultracentrifugation. Titers of all AAV vectors used for in vivo study were quantitated by quantitative reverse transcriptase-PCR. Given that the AAV8 vector is naturally mouse hepatocyte trophic, the AAT promoter is highly active in hepatocytes, and hepatocytes have a slow turnover, this approach results in sustained hepatocyte-selective expression of murine *Fgf21* in the long term. Since the recombinant AAV8 vector was generated by a standard and helper-free 3 plasmid transfection system, this vector does not express AAV8 and adenoviral helper proteins, and cannot replicate in transduced hepatocytes. Pilot data in C57BL/6 mice showed that the AAV8-*Fgf21* vector ($3×10^{10}$, $1×10^{11}$, and $1×10^{12}$ genome copies/mouse) did not cause liver injury, as judged from unaffected alanine transaminase and aspartate transaminase levels in plasma at 8 days after injection.

#### Body weight and plasma glucose, TG and TC

Body weight (n=18 per group) was recorded weekly of all mice throughout the study. Every 4 weeks, mice were fasted for 4 hr (9.00–13.00), and subsequently, tail vein blood was collected into paraoxon-coated glass capillaries. Plasma (n=18 per group) was collected and measured for glucose, TG, and TC using commercial enzymatic kits (Roche Diagnostics, Mannheim, Germany).

#### Plasma FGF21, adiponectin, and lipoprotein profile

Plasma FGF21 concentrations were determined at week –1 (pooled samples, n=6 per group), week 4 (pooled samples, n=6 per group), and week 23 (n=12–16 per group) using Mouse/Rat FGF21 Quantikine ELISA Kit (R&D Systems, Minneapolis, NE, USA). Plasma adiponectin levels were measured at week 22 (n=10 per group) using Mouse Adiponectin/Acrp30 Quantikine ELISA Kit (R&D Systems, Minneapolis, NE, USA). At week 22, 4 µL of 4 hr fasting plasma per mouse (n=18 per group) were pooled in each treatment group to measure the distribution of TG and TC over lipoproteins by fast-performance liquid chromatography using Super 6 column (GE Healthcare, Piscataway, NJ, USA).

#### Glucose tolerance test and lipid tolerance test

At week 16, an IPGTT was performed with an injection of D-glucose (2 g/kg body weight) after 4 hr fasting (9.00–13.00; n=8 per group). Blood was collected via tail vein at 0, 5, 15, 30, 60, and 120 min for each test. The glucose was measured with a OneTouch Ultra glucometer (AccuCheck Sensor, Roche Diagnostics, Almere, The Netherlands), and the area under the curve was calculated. During IPGTT, extra blood was collected at 0 and 15 min, spun down, and the serum samples were stored at –20°C for glucose measurement using a commercial enzymatic kit (Roche Diagnostics; Mannheim, Germany) and insulin measurement using an Ultra Sensitive Mouse Insulin ELISA kit (Crystal Chem, Zaandam, The Netherlands). HOMA-IR (homeostasis model assessment of insulin resistance) was calculated with the following formula: [fasting serum glucose (mM)×fasting serum insulin (µU/mL)]/22.5 (*Fraulob et al., 2010*). At week 20, oral lipid tolerance test was conducted. To this end, mice (n=10 per group) were fasted for 4 hr (9.00–13.00 hr), and received olive oil (10 mL/kg body weight) via oral gavage. Blood was collected into paraoxon-coated glass capillaries at 0, 2, 4, 6, and 8 hr, spun down, and the plasma samples were stored at –20°C for TG measurement using commercial enzymatic kits (Roche Diagnostics, Mannheim, Germany).

## Hepatic lipids and hydroxyproline

Hepatic lipids were extracted from snap-frozen liver samples (n=18 per group) using a modified protocol from *Bligh and Dyer, 1959*. Liver TG, TC and phospholipid (Instruchemie, Delfzijl, The Netherlands) and protein (Pierce, Thermo Fisher Scientific, Waltham, MA, USA) concentrations were measured. Hepatic lipids were expressed as nmol per mg protein. Hepatic hydroxyproline concentrations (n=18 per group) were determined using a Mouse Hydroxyproline Assay Kit (QuickZyme Biosciences, Leiden, The Netherlands).

## Adipose tissue histology

Formalin-fixed paraffin-embedded iBAT, sWAT, and gWAT sections (5 µm thickness) were prepared for hematoxylin-eosin (H&E) staining (*Cardiff et al., 2014*). Moreover, iBAT and sWAT sections were processed for UCP-1 staining (*Kooijman et al., 2015*), and gWAT sections were used for F4/80 staining (*Lanthier et al., 2010*). The areas occupied by intracellular lipid vacuoles (n=18 per group) and UCP-1 (n=18 per group) were quantified using ImageJ software (version 1.52a; National Institutes of Health, Bethesda, MD, USA). Using ImageJ software, the size of adipocyte of gWAT (n=18 per group) and sWAT (n=18 per group) and the number of CLSs within the gWAT (n=18 per group) were assessed. The number of CLSs in the gWAT was expressed as the number of CLS per mm$^2$.

## Liver histology and histological grading of NAFLD activity score

Liver tissue (n=18 per group) was fixed, embedded, and sectioned (5 µm thickness) for H&E, Oil Red O (ORO), F4/80, Picrosirius Red, and COL1A1 staining. The number of CLS in the liver was counted using ImageJ software and expressed as the number of CLS per mm$^2$. In addition, hepatic collagen accumulation was evaluated by quantifying Picrosirius Red- and COL1A1-positive areas in the liver using ImageJ software. For NAFLD activity score determination, a clinically utilized scoring system was adapted for the current study based on liver section H&E staining (*Bedossa et al., 2012*). The scoring system is ranged from 0 to 7, and is evaluated semi-quantitatively through three criteria: steatosis (0–3), lobular inflammation (0–2), and hepatocellular ballooning (0–2). Values in figures for each staining present means of 6–9 different and randomly analyzed fields (~1.5 mm$^2$) of each mouse, and were used for statistical analysis.

## Gene expression analysis

Total RNA was isolated from snap-frozen tissues (n=18 per group for each tissue) using TriPure RNA Isolation Reagent (Roche Diagnostics, Mijdrecht, The Netherlands). Thereafter, complementary DNA for quantitative reverse transcriptase-PCR was generated using Moloney Murine Leukemia Virus Reverse Transcriptase (Promega, Leiden, The Netherlands). Then, mRNA expression was normalized to *Actb* and *Rplp0* mRNA levels and expressed as fold change compared with the LFCD group. The primer sequences are listed in *Appendix 1—table 1*.

**Appendix 1—table 1.** List of polymerase chain reaction primer sequences used in mRNA expression analysis.

| Gene | Forward primer (5′–3′) | Reverse primer (5′–3′) |
|------|------------------------|------------------------|
| *Abcg5* | GAGCTGCAGAGGATGATTGCT | AGCCACCCTGGTCTTGGA |
| *Acta2* | CCTGACGGGCAGGTGATC | ATGAAAGATGGCTGGAAGAGAGTCT |
| *Actb* | AACCGTGAAAAGATGACCCAGAT | CACAGCCTGGATGGCTACGTA |
| *Adgre1* | CTTTGGCTATGGGCTTCCAGTC | GCAAGGAGGACAGAGTTTATCGTG |
| *Adipoq* | CTCCACCCAAGGGAACTTGT | TAGGACCAAGAAGACCTGCATC |
| *Apob* | GCCCATTGTGGACAAGTTGATC | CCAGGACTTGGAGGTCTTGGA |
| *Ccl2* | GCATCTGCCCTAAGGTCTTCA | TTCACTGTCACACTGGTCACTCCTA |
| *Cd36* | GCAAAGAACAGCAGCAAAATC | CAGTGAAGGCTCAAAGATGG |
| *Col1a1* | GAGAGAGCATGACCGATGGATT | TGTAGGCTACGCTGTTCTTGCA |
| *Cpt1a* | GAGACTTCCAACGCATGACA | ATGGGTTGGGGTGATGTAGA |

*Appendix 1—table 1 Continued on next page*

*Appendix 1—table 1 Continued*

| Gene | Forward primer (5′–3′) | Reverse primer (5′–3′) |
|---|---|---|
| *Ctgf* | GGCCTCTTCTGCGATTTCG | CCATCTTTGGCAGTGCACACT |
| *Cyp7a1* | CAGGGAGATGCTCTGTGTTCA | AGGCATACATCCCTTCCGTGA |
| *Cyp8b1* | GGACAGCCTATCCTTGGTGA | CGGAACTTCCTGAACAGCTC |
| *Cyp27a1* | TCTGGCTACCTGCACTTCCT | CTGGATCTCTGGGCTCTTTG |
| *Codon-optimized Fgf21* | GCCCACCTGGAGATCAGGGAGGA | GGCAGGAAGCGCACAGGTCCCCAG |
| *Fgf21* | GGGGTCATTCAAATCCTGGGTGTCA | ACACATTGTAACCGTCCTCCAGCAG |
| *Fgfr1* | AGAGTCCAAGAGTAAAAGCAGC | CTTCCGAGGTTCAGCTCTCC |
| *Fgfr2* | GCTATAAGGTACGAAACCAGCAC | GGTTGATGGACCCGTATTCATTC |
| *Fgfr4* | TCCATGACCGTCGTACACAAT | ATTTGACAGTATTCCCGGCAG |
| *Il1b* | GCAACTGTTCCTGAACTCAACT | ATCTTTTGGGGTCCGTCAACT |
| *Klb* | TGTTCTGCTGCGAGCTGTTAC | TACCGGACTCACGTACTGTTT |
| *Mttp* | CTCTTGGCAGTGCTTTTTCTCT | GAGCTTGTATAGCCGCTCATT |
| *Pgc1a* | TGCTAGCGGTTCTCACAGAG | AGTGCTAAGACCGCTGCATT |
| *Ppara* | ATGCCAGTACTGCCGTTTTC | GGCCTTGACCTTGTTCATGT |
| *Rplp0* | GGACCCGAGAAGACCTCCTT | GCACATCACTCAGAATTTCAATGG |
| *Tgfb1* | TTGCCCTCTACAACCAACACAA | GGCTTGCGACCCACGTAGTA |
| *Tnfa* | AGCCCACGTCGTAGCAAACCAC | TCGGGGCAGCCTTGTCCCTT |

Abcg5, ATP-binding cassette transporter G member 5; *Acta2*, actin α2; *Actb*, β-actin *Adgre1*, adhesion G protein-coupled receptor E1; *Adipoq*, adiponectin; *Apob*, apolipoprotein B; *Ccl2*, C–C motif chemokine ligand 2; *Cd36*, cluster of differentiation 36; *Col1a1*, collagen type 1α1; *Cpt1a*, carnitine palmitoyl transferase 1α; *Ctgf*, connective tissue growth factor; *Cyp7a1*, cholesterol 7α-hydroxylase; *Cyp8b1*, sterol 12α-hydroxylase; *Cyp27a1*, sterol 27-hydroxylase; *Fgf21*, exogenous fibroblast growth factor; *Fgfr*, fibroblast growth factor receptor; *Il1b*, interleukin-1β; *Klb*, β-Klotho; *Pgc1a*, peroxisome proliferator-activated receptor gamma coactivator 1α; *Ppara*, peroxisome proliferator-activated receptor α; *Rplp0*, ribosomal protein lateral stalk subunit p0; *Tgfb1*, transforming growth factor-β; *Tnfa*, tumor necrosis factor α.

## Isolation of hepatic leukocytes

At the end of the study, livers (n=5 per group) were collected in ice-cold RPMI 1640+Glutamax (Thermo Fisher Scientific, Waltham, MA, USA). The tissues were subsequently minced and digested for 45 min at 37°C in RPMI 1640+Glutamax supplemented with 1 mg/mL collagenase type IV from *Clostridium histolyticum* (Sigma-Aldrich, St Louis, MO, USA), 2000 U/mL DNase (Sigma-Aldrich, St Louis, MO, USA) and 1 mM $CaCl_2$ as previously described (*van der Zande et al., 2021*). The digested liver tissues were passed through a 100 μm cell strainer and washed with PBS supplemented with 0.5% BSA and 2 mM EDTA (PBS/BSA/EDTA). The samples were spun down (530 × *g*, 10 min at 4°C) after which the pellet was resuspended in PBS/BSA/EDTA and centrifuged at 50 × *g* to pellet the hepatocytes (3 min at 4°C). The supernatant was next collected and centrifuged (530 × *g*, 10 min at 4°C) after which the pellet was treated with erythrocyte lysis buffer (0.15 M $NH_4Cl$; 1 mM $KHCO_3$; 0.1 mM $Na_2EDTA$) for 2 min at room temperature. After washing with PBS/BSA/EDTA, total leukocytes were isolated by means of magnetic-activated cell sorting (MACS) using LS columns and CD45 MicroBeads (35 μL beads per liver; Miltenyi Biotec, Bergisch Gladbach, Germany) according to the manufacturer's protocol. Isolated CD45+ cells were counted and stained with Zombie NIR (BioLegend, San Diego, CA, USA) for 20 min at room temperature followed by fixation with 1.9% paraformaldehyde (Sigma-Aldrich, St Louis, MO, USA) for 15 min at room temperature after which the fixed leukocytes were further processed for flow cytometry.

## Flow cytometry

For analysis of hepatic leukocyte subsets, isolated CD45+ cells were incubated with a cocktail of antibodies directed against XCR1, CD11c, CD19, Ly6G, F4/80, MHC-II, CD45, CLEC2, Siglec-F, CD64, CD8, NK1.1, CD11b, CD4, CD90.2, Ly6C, CD3, CD36, CD9, and TIM4 in PBS/BSA/EDTA supplemented with True-Stain monocyte blocker (BioLegend, San Diego, CA, USA) and Brilliant

Stain Buffer Plus (BD Biosciences, Franklin Lakes, NJ, USA) for 30 min at 4°C. The stained samples (n=5 per group) were measured by spectral flow cytometry using a Cytek Aurora spectral flow cytometer (Cytek Biosciences, Fremont, CA, USA). Spectral unmixing of the flow cytometry data was performed using SpectroFlo v3.0 (Cytek Biosciences, Fremont, CA, USA). Gating of flow cytometry data was performed using FlowJo v10.8 Software (BD Biosciences, Franklin Lakes, NJ, USA). Dimensionality reduction by means of Uniform Manifold Approximation and Projection (UMAP) was performed using OMIQ data analysis software (Omiq Inc, Santa Clara, CA, USA). Statistical analysis was performed using GraphPad version 9.01 for Windows (GraphPad Software, La Jolla, CA, USA). Representative gating strategies are shown in *Figure 5—figure supplement 2A* and information regarding the antibodies used is listed in *Appendix 1—table 2*.

**Appendix 1—table 2.** List of antibodies and other reagents used for flow cytometry analyses.

| Target | Clone | Conjugate | Source | Catalog number |
| --- | --- | --- | --- | --- |
| CD3 | 17A2 | APC/Fire-810 | BioLegend | 100267 |
| CD4 | RM4-5 | APC | eBioscience | 17-0042-83 |
| CD8 | RPA-T8 | PE-Cy5 | BD Biosciences | 561951 |
| CD9 | MZ3 | PerCP-Cy5.5 | BioLegend | 124817 |
| CD11b | M1/70 | PE-Cy7 | eBioscience | 25-0112-82 |
| CD11c | HL3 | V450 | BD Biosciences | 560521 |
| CD19 | 1D3 | BV480 | BD Biosciences | 566107 |
| CD36 | HM36 | PE | BioLegend | 102606 |
| CD45 | 30-F11 | BV785 | BioLegend | 103149 |
| CD64 | X54-5/7.1 | PE-DAZZLE594 | BioLegend | 139320 |
| CD90.2 | 30-H12 | Alexa Fluor 700 | BioLegend | 105319 |
| CLEC2 | 17D9 | FITC | Bio-Rad | MCA5700 |
| F4/80 | BM8 | BV711 | BioLegend | 123147 |
| Ly6C | HK1.4 | APC-Cy7 | BioLegend | 128025 |
| Ly6G | 1A8 | BV650 | BioLegend | 127641 |
| MHC-II | M5/114.15.2 | BV750 | BD Biosciences | 747458 |
| MHC-II | M5/114.15.2 | Alexa Fluor 700 | Thermo Fisher | 56-5321-82 |
| NK1.1 | PK136 | PerCP-Cy5.5 | BioLegend | 108727 |
| Siglec-F | E50-2440 | PE | BD Biosciences | 552126 |
| Siglec-F | E50-2440 | BV605 | BD Biosciences | 740388 |
| TIM4 | 54 (RMT4-54) | PerCP-eFluor710 | Thermo Fisher | 46-5866-82 |
| XCR1 | ZET | BV421 | BioLegend | 148216 |
| **Other reagents** | | | | |
| Zombie NIR Fixable Viability Kit | | | BioLegend | 423106 |
| True-Stain Monocyte Blocker | | | BioLegend | 426103 |
| Brilliant Stain Buffer Plus | | | BD Biosciences | 566385 |

