## [Editor Report]

The study examines the mechanism of hepatic FGF21 using trangenic and over-expression models to show that it limits hepatic lipotoxicity, inflammation and fibrogenesis. They provide convincing data to support the notion that FGF21 blocks hepatic lipid influx and accumulation through combined endocrine and autocrine signaling, respectively, which prevent Kupffer cell activation, and scar-associated macrophages to inhibit fibrogenesis.

---

## [Decision Letter]

**Decision letter after peer review:**

Thank you for submitting your article "FGF21 protects against hepatic lipotoxicity and macrophage activation to attenuate fibrogenesis in nonalcoholic steatohepatitis" for consideration by *eLife*. Your article has been reviewed by 2 peer reviewers, and the evaluation has been overseen by a Reviewing Editor and Mone Zaidi as the Senior Editor. The reviewers have opted to remain anonymous.

Essential revisions:

1) More information on the AAV construct is needed.

a. Is the sequence human or murine FGF21?

b. Does AAV mediate FGF21 expression in other liver cells, for instance endothelial or Kupffer cells?

c. What happens to the endogenous FGF21 mRNA upon the viral gain of function?

d. AAV usually allows an expression for up to 21-28 days; how did the authors manage to have such long persistence of the mRNA and protein expression?

e. Is this construct able to replicate in vivo?

f. Were serum ALT/AST sampled acutely post injections (time-course) during the pilot to find the best compromise between expression and cytotoxicity?

g. Any limitations of the AAV model should be discussed

2) Please provide a rationale as to why the 23-week time point was chosen, as it appears the FGF21 effect on weight and cardiometabolic parameters may have already occurred within the first few weeks of FGF21 overexpression. Also, were FGF21 levels assayed at any point between 4 weeks and 23 weeks?

3) FGF21 is considered a stress-induced hormone whose levels rise in situations of metabolic stress including diet-induced obesity and NASH. As such, one would expect to see some upregulation (even if mild) of FGF21 in HFCD mice (Figure 1A). How would the authors explain the lack of FGF21 upregulation?

4) Clarify the methods for histological analysis of the liver in Figure 4. It seems to be based on the SAF criteria, which should take discrete values of 0,1,2,3 for steatosis, 0,1,2 for inflammation, and 0,1,2 for ballooning. However, particularly for inflammation and ballooning, the values are clearly graded along the scale.

5) Given the differences in brown fat activity and physiology between mice and humans, it would be important for the authors to either moderate their comments on the UCP1 dependence of their phenotype or provide more data to clarify to what extent their findings are UCP1-dependent (e.g. food intake in the FGF21 overexpression model and/or evidence of increased energy expenditure).

---

## [Author Response]

Essential Revisions (for the authors):1) More information on the AAV construct is needed.a. Is the sequence human or murine FGF21?

The AAV8 vector used in the present study expresses codon-optimized murine *Fgf21* cDNA under the control of a strong liver-specific apolipoprotein E (*Apoe*)/α-1-antitrypsin (*Aat*) promoter. We have included the following text in the revised manuscript.

Materials and methods

Page 20 Line 450: “… expressing codon-optimized murine *Fgf21* …”

Supporting Material and Methods

Page 2 Lines 35-36: “…and either pAAV apolipoprotein E (*Apoe*)/α-1-antitrypsin (*Aa*t) promoter-driven codon-optimized murine *Fgf21* or…”

b. Does AAV mediate FGF21 expression in other liver cells, for instance endothelial or Kupffer cells?

As mentioned in our response to a, we used an AAV8-*Apoe*/*Aat* vector in this study. Given that the AAV8 vector is naturally mouse hepatocyte trophic, and *Aat* promoter is highly active in hepatocytes, this approach should result in hepatocyte-selective expression of murine FGF21. We have now included the following text (which also includes responses to other questions) in the Supporting Materials and methods.

Page 2 Lines 40-43: “Given that the AAV8 vector is naturally mouse hepatocyte trophic, the AAT promoter is highly active in hepatocytes, and hepatocytes have a slow turnover, this approach results in sustained hepatocyte-selective expression of murine *Fgf21* in the long-term.”

c. What happens to the endogenous FGF21 mRNA upon the viral gain of function?

Following the editors and reviewers’ comment on endogenous FGF21 mRNA expression, we have now measured hepatic endogenous *Fgf21* expression. We observed that HFCD feeding increased endogenous *Fgf21* expression, which was prevented by viral overexpression of codon-optimized murine FGF21 (new Figure 1B).

The new data and related descriptions have been included in the revised manuscript.

Results

Page 8 Lines 149-151: “In addition, we observed that HFCD feeding increased hepatic endogenous *Fgf21* expression (+184%), which, however, was prevented by AAV8-*Fgf21* administration (Figure 1B).”

d. AAV usually allows an expression for up to 21-28 days; how did the authors manage to have such long persistence of the mRNA and protein expression?

In non-dividing or slowly dividing cell types such as hepatocytes, AAV8 vectors do allow long-term transgene expression. In fact, upon liver-targeted AAV8-luciferase delivery to mice, we have seen sustained luciferase expression at least for 1 year. In fact, in a human hemophilia B gene therapy trial, AAV8 vector-mediated delivery of Factor IX resulted in stable Factor IX expression over a period of 8 years following systemic administration in patients [1]. We have now included the following text in the Supporting Materials and methods.

Page 2 Lines 42-43: “… hepatocytes have slow turnover, this approach results in sustained hepatocyte-selective expression of murine *Fgf21* in the long-term.”

e. Is this construct able to replicate in vivo?

The recombinant AAV8 vector was generated by a standard and helper-free 3 plasmid transfection system. Therefore, this vector does not express AAV8/adenoviral helper proteins, and cannot replicate in transduced hepatocytes. We have now added the following text in the Supporting Materials and methods.

Page 2 Lines 43-46: “Since the recombinant AAV8 vector was generated by a standard and helper-free 3 plasmid transfection system, this vector does not express AAV8/adenoviral helper proteins, and cannot replicate in transduced hepatocytes.”

f. Were serum ALT/AST sampled acutely post injections (time-course) during the pilot to find the best compromise between expression and cytotoxicity?

Before conducting the present study, we tested the acute effect of AAV8-FGF21 on plasma ALT and AST levels in high-fat diet (HFD)-fed wild-type mice after 8-days post the vector administration (see Author response image 1). As expected, plasma ALT and AST levels in mice of the HFD+PBS group were elevated when compared to those in the Chow group. However, as compared to the HFD+PBS group, we did not observe acute increases in plasma ALT and AST levels in mice of HFD+AAV8-FGF21 groups.

**Author response image 1. sa2fig1:** Acute effects of AAV8-FGF21 administration on plasma ALT/AST levels in mice. C57BL/6 mice were fed a high-fat diet (HFD) for 40 weeks, after which these mice (n = 4-5) were treated with either PBS or various doses of AAV8-FGF21 vectors (3 × 10^10^, 1 × 10^11^ and 1 × 10^12^ genome copies/mouse). After 8 days of AAV8-FGF21 interventions, these mice were terminated, and plasma samples were collected for ALT and AST measurement. Data are shown as mean ± SD. Differences were assessed using one-way ANOVA followed by a Tukey post-test. ****P < 0.0001, compared with the Chow group.

Therefore, these data indicate that the used dose of the vector in the present study does not induce acute liver injury.

We have now added the following text in the Supporting Materials and methods.

Page 2 Lines 46-49: “ Pilot data in C57BL/6 mice showed that the AAV8-*Fgf21* vector (3x10^10^, 1x10^11^ and 1x10^12^ genome copies/mouse) did not cause liver injury, as judged from unaffected alanine transaminase (ALT) and aspartate transaminase (AST) levels in plasma at 8 days after injection.”

g. Any limitations of the AAV model should be discussed

Following editors’ and reviewers’ comments, we have included the following text in the Discussion.

Page 18-19 Lines 417-427: “This study is not without limitations. In this work, we used a gene therapy approach to examine the effects of liver-derived FGF21 on NASH based on the use of a single injection of an AAV8 vector encoding codon-optimized murine FGF21. Although AAV8 is hepatocyte trophic, we have not excluded potential contribution of other hepatic cells to total FGF21 expression. Also, while AAV8-*Fgf21* was non-toxic, sustained supra-pharmacological plasma levels of FGF21 were achieved, which do not necessarily reflect effects of current pharmacological strategies with long-acting FGF21. Interestingly, AAV-mediated gene therapy has already been tested in the clinic for life-threatening diseases such as hemophilia B, and has demonstrated stable expression of factor IX following AAV-mediated delivery (59). Therefore, it is reasonable to speculate that liver-targeted gene therapy as an approach to induce stable overexpression of FGF21 may ultimately have potential to reach to the clinic.”

2) Please provide a rationale as to why the 23-week time point was chosen, as it appears the FGF21 effect on weight and cardiometabolic parameters may have already occurred within the first few weeks of FGF21 overexpression. Also, were FGF21 levels assayed at any point between 4 weeks and 23 weeks?

In the present study, we aimed to investigate the role of FGF21 in hepatic inflammation and early fibrosis in NASH development. Based on previous studies [2, 3] and a well-established protocol in our research group, we observed that upon the 23-week HFCD dietary intervention, APOE*3-Leiden.CETP mice exhibit hepatic inflammation and early fibrosis. We have now included the following text in the Materials and methods.

Page 20 Line 456-457: “…, at which APOE*3-Leiden.CETP mice have developed both steatosis, hepatic inflammation and early fibrosis (23, 61).”

As mentioned in (1), the AAV8 vector used here induces a long-lasting increase in FGF21 levels in the circulation. We therefore did not measure plasma FGF21 levels between weeks 4 and 23.

3) FGF21 is considered a stress-induced hormone whose levels rise in situations of metabolic stress including diet-induced obesity and NASH. As such, one would expect to see some upregulation (even if mild) of FGF21 in HFCD mice (Figure 1A). How would the authors explain the lack of FGF21 upregulation?

Although significant differences were not achieved using one-way ANOVA followed by a Tukey post-test, by performing a student *t*-test between the LFCD and HFCD groups, we did observe higher plasma FGF21 levels in the HFCD group compared to those of the LFCD group (see Author response image 2). We have also measured hepatic endogenous *Fgf21* expression (see response to comment 1c) to show that HFCD feeding did significantly increase the hepatic expression of endogenous *Fgf21*. Moreover, we showed that hepatic endogenous FGF21 expression was positively correlated with plasma endogenous FGF21 levels (see Author response image 2). Therefore, our data in fact are in agreement with previous studies showing increased FGF21 levels in patients with NASH and its associated metabolic disorders [4, 5].

**Author response image 2. sa2fig2:** HFCD increases endogenous FGF21 levels. (A) Plasma endogenous FGF21 levels were measured before (at week -1; pooled samples, n = 6 per group) and after (at week 4, pooled samples, n = 6 per group; week 23, n = 12-16 per group) AAV8-null administration. (B) Plasma endogenous FGF21 levels (week 23) were plotted against the expression of hepatic endogenous FGF21 expression. Data are shown as mean ± SEM. (A) Differences were assessed using student t-test.(B) Linear regression analyses were performed.**P < 0.01, compared with the LFCD group.

Results

Page 8 Line 151-156: “Furthermore, by performing a student *t*-test between the LFCD and HFCD groups, we did observe that as compared to the LFCD group, HFCD feeding increased plasma FGF21 levels at week 4 (+52%) and week 23 (+383%) (Figure 1C).These results are in agreement with previous findings showing that FGF21 is a stress-induced hepatokine whose levels increase in metabolically compromised states, such as obesity (16) and NAFLD (17).”

4) Clarify the methods for histological analysis of the liver in Figure 4. It seems to be based on the SAF criteria, which should take discrete values of 0,1,2,3 for steatosis, 0,1,2 for inflammation, and 0,1,2 for ballooning. However, particularly for inflammation and ballooning, the values are clearly graded along the scale.

We apologize this was not sufficiently clear. We have now included the following text in the Supporting Materials and methods.

Page 5 Lines 114-116: “Values in figures for each staining present means of 6-9 different and randomly analyzed fields (~1.5 mm^2^) of each mouse, and were used for statistical analysis.”

5) Given the differences in brown fat activity and physiology between mice and humans, it would be important for the authors to either moderate their comments on the UCP1 dependence of their phenotype or provide more data to clarify to what extent their findings are UCP1-dependent (e.g. food intake in the FGF21 overexpression model and/or evidence of increased energy expenditure).

Following editors’ and reviewers’ comments, we have edited and added the following text in the Discussion.

Page 16 Lines 355-360: “These data imply that FGF21 induces thermogenesis which increases energy expenditure, consistent with the thermogenic responses observed for recombinant FGF21 in C57BL/6 mice fed with an obesogenic diet (29). Likewise, by using APOE*3-Leiden.CETP mice, we previously reported that FGF21 treatment highly increased energy expenditure without affecting food intake (30).”

References:

1. Nathwani, A.C., et al., Adeno-Associated Mediated Gene Transfer for Hemophilia B:8 Year Follow up and Impact of Removing "Empty Viral Particles" on Safety and Efficacy of Gene Transfer. Blood, 2018. 132.

2. Morrison, M.C., et al., Mirtoselect, an anthocyanin-rich bilberry extract, attenuates non-alcoholic steatohepatitis and associated fibrosis in ApoE( *)3Leiden mice. J Hepatol, 2015. 62(5): p. 1180-6.

3. Hui, S.T., et al., The Genetic Architecture of Diet-Induced Hepatic Fibrosis in Mice. Hepatology, 2018. 68(6): p. 2182-2196.

4. Li, H., et al., Fibroblast growth factor 21 levels are increased in nonalcoholic fatty liver disease patients and are correlated with hepatic triglyceride. J Hepatol, 2010. 53(5): p. 934-40.

5. Dushay, J., et al., Increased fibroblast growth factor 21 in obesity and nonalcoholic fatty liver disease. Gastroenterology, 2010. 139(2): p. 456-63.